# Semi-off-Policy Reinforcement Learning for Vision-Language Slow-Thinking Reasoning

Junhao Shen[1,2]    Haiteng Zhao[2]    Yuzhe Gu[1,2]    Songyang Gao[2]    Kuikun Liu[2]    Haian Huang[2]

Jianfei Gao[2]    Dahua Lin[2,3]    Wenwei Zhang[2†]    Kai Chen[2†]

[1]Shanghai Jiao Tong University    [2]Shanghai AI Laboratory
[3]MMLab, The Chinese University of Hong Kong
{shenjunhao,zhangwenwei,chenkai}@pjlab.org.cn

## Abstract

Enhancing large vision-language models (LVLMs) with visual slow-thinking reasoning is crucial for solving complex multimodal tasks. However, since LVLMs are mainly trained with vision-language alignment, it is difficult to adopt on-policy reinforcement learning (RL) to develop the slow thinking ability because the rollout space is restricted by its initial abilities. Off-policy RL offers a way to go beyond the current policy, but directly distilling trajectories from external models may cause visual hallucinations due to mismatched visual perception abilities across models. To address these issues, this paper proposes **SOPHIA**, a simple and scalable **S**emi-**O**ff-**P**olicy RL for vision-language slow-t**HI**nking re**A**soning. SOPHIA builds a semi-off-policy behavior model by combining on-policy visual understanding from a trainable LVLM with off-policy slow-thinking reasoning from a language model, assigns outcome-based rewards to reasoning, and propagates visual rewards backward. Then LVLM learns slow-thinking reasoning ability from the obtained reasoning trajectories using propagated rewards via off-policy RL algorithms. Extensive experiments with InternVL2.5 and InternVL3.0 with 8B and 38B sizes show the effectiveness of SOPHIA. Notably, SOPHIA improves InternVL3.0-38B by 8.50% in average, reaching state-of-the-art performance among open-source LVLMs on multiple multimodal reasoning benchmarks, and even outperforms some closed-source models (*e.g.*, GPT-4.1) on the challenging MathVision and OlympiadBench, achieving 49.08% and 49.95% pass@1 accuracy, respectively. Analysis shows SOPHIA outperforms supervised fine-tuning and direct on-policy RL methods, offering a better policy initialization for further on-policy training.

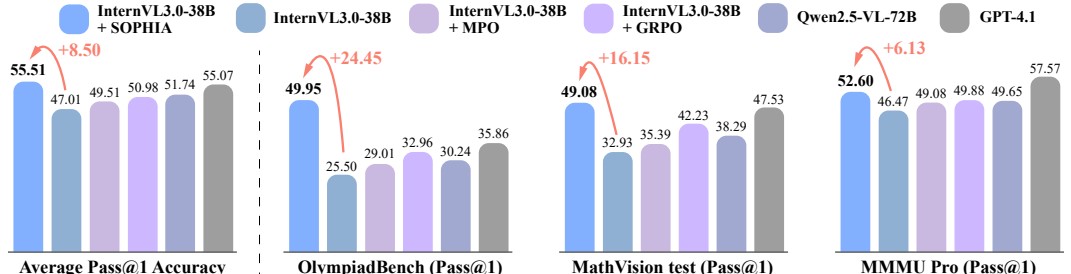

Figure 1: The average pass@1 accuracy of SOPHIA and some representative models across eight benchmarks, as well as their pass@1 accuracy on OlympiadBench, MathVision test, and MMMU Pro. InternVL3.0-38B + SOPHIA significantly improves over its base model, InternVL3.0-38B (Instruct), achieving state-of-the-art performance on most of the benchmarks.

39th Conference on Neural Information Processing Systems (NeurIPS 2025).

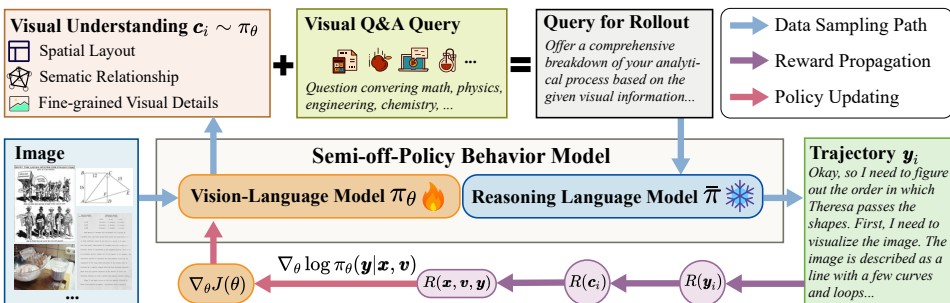

Figure 2: An overview of SOPHIA. SOPHIA samples reasoning trajectories $y_i$ and calculates rewards for both reasoning $R(y_i)$ and visual understanding $R(c_i)$. Finally, it updates the LVLM policy via an off-policy method to cultivate visual slow-thinking reasoning abilities.

# 1 Introduction

Leveraging reasoning abilities to solve complex problems is a crucial step toward artificial general intelligence [1, 2]. Recent large language models (LLMs), *e.g.*, OpenAI o-series [3] and DeepSeek-R1 [4], have excelled at complex problem solving by their slow thinking capabilities [5, 6]. These advances inspire the community to explore large vision-language models (LVLMs) for complex multimodal problem solving (*e.g.*, geometry problems) with visual slow thinking [7–10].

Early explorations mainly improve the reasoning capability of LVLMs through pipeline-generated reasoning trajectories [11–14] or distillation [15], yet, these methods lead to pattern memorization rather than genuine improvement in visual reasoning [16]. Inspired by the recent success of reinforcement learning (RL) for LLMs reasoning, especially DeepSeek-R1 [4], recent works explore on-policy algorithms (*e.g.*, GRPO [17]) in LVLMs and try to elicit the reflection behavior in training [18–21]. However, since most LVLMs are trained through vision-text alignment data [22, 23] that lack slow-thinking reasoning trajectories, it is hard to sample slow-thinking behaviors from the output space of LVLMs. Consequently, the performance of on-policy RL are fundamentally bounded by the initial policy distribution, as they essentially only amplify existing behaviors inside LVLMs [24, 25].

Given the limitation of on-policy RL, off-policy learning offers a promising solution. By using alternative policy models to generate trajectories, it bypasses the limitations of the current policy distribution and enables learning beyond their initial abilities [26, 27]. However, the visual features involved in off-policy slow-thinking reasoning trajectories may not be aligned with the own visual understanding abilities of LVLMs, leading to a conflict between the optimization direction and the visual understanding of LVLMs [28–30], which severely exacerbates issues like hallucinations in LVLM visual slow-thinking reasoning [31, 32].

To overcome the limitations of both on-policy and off-policy RL for LVLMs reasoning, this paper proposes **SOPHIA**, **S**emi-**O**ff-**P**olicy RL for vision-language slow-t**HI**nking re**A**soning. As shown in Figure 2, SOPHIA consists of two steps: semi-off-policy sampling and policy optimization with propagated rewards. Specifically, SOPHIA builds a semi-off-policy behavior model by combining the on-policy visual understanding from the trainable LVLM with the off-policy slow thinking from an open-source reasoning LLMs (*e.g.*, QwQ [33]). During trajectory collection, SOPHIA first samples a batch of image understandings (*i.e.*, detailed descriptions of images) from LVLMs for the same query, and then adopts reasoning models for sampling slow-thinking reasoning trajectories.

In the policy optimization process, SOPHIA first calculates rewards for the slow-thinking reasoning trajectories of LLMs based on the verification results of the reasoning outcome. Because outcome-based rewards may cause the model to overlook the accuracy and completeness of visual content, we assign each visual understanding a reward via reward backpropagation to strengthen the link between visual understanding and reasoning. Such a design not only enhances the slow-thinking abilities of LVLM but also optimizes the use of visual understanding in reasoning with visual rewards. Furthermore, SOPHIA is scalable as it uses the propagation of outcome-based rewards instead of human or closed-source model annotations for ensuring the quality of visual understanding.

Extensive experiments show that SOPHIA effectively enhances the ability of performing slow thinking on challenging visual tasks. Experiments on both InternVL2.5 and InternVL3.0 at 8B and 38B show that SOPHIA improves performance across various benchmarks. Specifically, InternVL3.0-38B +

SOPHIA reaches state-of-the-art results on several benchmarks, which achieves an average gain of +8.50% and attains 49.08% and 49.95% pass@1 accuracy on the challenging MathVision [34] and OlympiadBench [35] datasets, outperforming leading open-source models such as Qwen2.5-VL-72B [36], as well as closed-source models like GPT-4.1 [37]. Further results show that SOPHIA outperforms both supervised fine-tuning and on-policy RL methods. Continued on-policy RL training with SOPHIA leads to substantial improvements, achieving performance comparable to Claude-3.7-Sonnet on general college-level questions. Moreover, in-depth analysis and ablation studies including diverse settings and training data further verify the generality and scalability of SOPHIA.

## 2 Related Work

**Visual Slow-thinking Reasoning Tasks.** Large vision-language models (LVLMs) are thriving in various visual tasks. Both closed-source models (*e.g.*, ChatGPT 4o [38] and Claude 3.7 Sonnet [39]) and open-source models (*e.g.*, QwenVL [36] and InternVL [40]) have made key progress in image captioning [41], object detection [42] and video understanding [43]. Recent efforts focus on complex visual reasoning tasks which require both slow-thinking and visual understanding [7–9]. However, it remains challenging to equip LVLMs with robust slow-thinking reasoning. Although Chain-of-Thought (CoT) techniques enhance reasoning of LLMs [44], applying them directly to LVLMs often fails to improve the textual and visual reasoning [45, 46]. This is largely because most LVLMs rely on vision-text alignment during pre-training and fine-tuning [22, 23], which hinders the development of genuine visual slow-thinking reasoning.

**Supervised Fine-tuning for LVLM Reasoning.** Previous efforts to endow LVLMs with reasoning abilities often rely on supervised approaches which involve constructing reasoning trajectories or structured pipelines to generate training data. For instance, M$^3$CoT [11] constructs CoT using a multi-step workflow, and Mulberry [14] employs Collective Monte Carlo Tree Search to generate CoT. More recently, large datasets are synthesized by leveraging LLMs to generate visually-informed reasoning trajectories based on image captions [15]. However, generated data often includes visual hallucinations [47] and these pipelines rarely consider it. Training on such noisy data may degrade the performance [48]. Additionally, supervised fine-tuning tends to encourage models to memorize specific reasoning patterns rather than develop robust and generalizable visual slow-thinking reasoning capabilities [16]. Compared to these methods, SOPHIA offers better scalability and generalization.

**Reinforcement Learning for LVLM Reasoning.** Reinforcement learning (RL) offers better generalization and is regarded as a more fundamental and effective training approach [16]. Extensive research has explored the use of on-policy RL to enhance slow-thinking reasoning abilities, *e.g.*, RLOO [49, 50], PPO [51], GRPO [17] and DAPO [52]. Recently, inspired by the success of LLMs, some studies apply on-policy RL to LVLMs. MM-Eureka [20] and Vision-R1 [19] mainly leverage GRPO to improve the reasoning abilities of LVLM. However, on-policy methods are fundamentally limited by existing policy distributions of LVLM, reinforcing suboptimal behaviors and hindering further progress. While off-policy RL is promising, its application to LVLMs remains challenging: the visual features in off-policy slow-thinking trajectories may misalign with the inherent visual understanding of LVLM, creating optimization conflicts that exacerbate hallucinations in visual reasoning [28–32]. Compared to these methods, SOPHIA enables the model to better perceive visual features from trajectories and more effectively leverage its visual understanding during reasoning.

## 3 Preliminaries

**Policy Model.** Causal large vision-language model (LVLM) and large language model (LLM) are adopted to produce the trajectory. A causal vision-language models produces an output trajectory $\boldsymbol{y} = (y_1, y_2, \cdots, y_{L_y})$ given a textual input prompt $\boldsymbol{x} = (x_1, x_2, \cdots, x_{L_x})$ and image input $\boldsymbol{v} = (v_1, v_2, \cdots, v_{L_v})$, where $L_y, L_x, L_v$ represent the length of the token sequence, and each entry of these sequences belong to the token space of LVLM. A language generation policy $\pi$ in an auto-regressive model is characterized by a conditional probability distribution parameterized by $\theta$ as

$$\pi_\theta(\boldsymbol{y}|\boldsymbol{x}, \boldsymbol{v}) = \prod_{l=1}^{L_y} \pi_\theta(y_l|\boldsymbol{y}_{1:l-1}, \boldsymbol{x}, \boldsymbol{v}), \tag{1}$$

with the convention $\boldsymbol{y}_{1:0} = \emptyset$ and $\boldsymbol{y}_{1:l-1} = (y_1, y_2, \cdots, y_{l-1})$. Similarly, a causal LLM $\bar{\pi}$ is

$$\bar{\pi}(\boldsymbol{y}|\boldsymbol{x}) = \prod_{l=1}^{L_y} \bar{\pi}(y_i|\boldsymbol{y}_{1:l-1}, \boldsymbol{x}). \tag{2}$$

**Off-policy Optimization.** In our method, we focus on off-policy settings [53] and adopt policy gradient methods based on importance sampling [54, 55], which maximizes the expected cumulative returns. The objective function is formulated as

$$J(\theta) \triangleq \mathbb{E}_{(\boldsymbol{x},\boldsymbol{v},\boldsymbol{y})\sim\mathcal{D}} \left[ \frac{\pi_\theta(\boldsymbol{y}|\boldsymbol{x},\boldsymbol{v})}{\mu(\boldsymbol{y}|\boldsymbol{x},\boldsymbol{v})} R(\boldsymbol{x},\boldsymbol{v},\boldsymbol{y}) \right], \tag{3}$$

where $\mathcal{D}$ is the off-policy dataset containing samples $(\boldsymbol{x}, \boldsymbol{v}, \boldsymbol{y})$ along with their rewards $R(\boldsymbol{x}, \boldsymbol{v}, \boldsymbol{y})$, $\pi_\theta$ is the policy model (*i.e.*, LVLM) with learnable parameters $\theta$ and $\mu$ is the behavior policy that generates the offline data. Unlike on-policy RL, off-policy RL learns from trajectories generated by a different behavior policy, enabling more efficient exploration beyond the model's current abilities.

## 4 Semi-off-Policy Reinforcement Learning

An overview of SOPHIA is shown Figure 2. In the first stage (Section 4.1), SOPHIA builds a semi-off-policy behavior model by combining the on-policy visual understanding from the trainable large vision language model (LVLM) with the off-policy slow thinking from an open-source reasoning LLM, which is used to sample a number of image descriptions and description-derived reasoning trajectories. Next (Section 4.2), SOPHIA assigns outcome-based rewards to reasoning and propagates visual rewards backward. Finally (Section 4.3), we use off-policy RL to update the LVLM, with theoretical analysis showing that importance sampling involving the semi-off-policy behavior can be efficiently approximated via our reasoning and visual rewards.

### 4.1 Semi-off-policy Sampling

We construct a semi-off-policy behavior model by combining on-policy visual understanding from the trainable LVLM $\pi_\theta$ and off-policy slow-thinking reasoning from an open-source language model $\bar{\pi}$. Other existing multimodal models (*e.g.*, GPT-4o) are not employed because current LVLMs have limited visual understanding. As a result, visual features involved in off-policy reasoning trajectories may not be fully perceived or understood by the target LVLM. Thus, directly using trajectories containing visual features extracted from other behavior LVLMs can lead to conflicts between the gradient direction and the LVLM's inherent visual understanding biases. Even when these reasoning trajectories yield correct results, they still face a bottleneck during training.

Specifically, given a visual training dataset denoted as $\mathcal{D}_{\text{train}} = \{(I_i, \boldsymbol{x}_{q_i}, a_i)|i = 1, 2, \cdots, |\mathcal{D}_{\text{train}}|\}$, where $I_i$ is the $i$-th image, $\boldsymbol{x}_{q_i}$ is the $i$-th query, $a_i$ is the $i$-th golden answer to $\boldsymbol{x}_{q_i}$ and $|\cdot|$ calculates the size of set, the LVLM first performs visual understanding in an on-policy manner. For each image $I_i$, we first extract its visual tokens $\boldsymbol{v}_i = V(I_i)$ using the pretrained visual encoder $V$ inside the LVLM. Then, prompt engineering [56] making the LVLM try its best to capture spatial layouts, semantic relationships among objects, fine-grained visual details and so on (Details in Figure A2 of Appendix E). Then, we can obtain a conditional distribution of image description $\boldsymbol{c}_i \sim \pi_\theta(\cdot|\boldsymbol{x}_d, \boldsymbol{v}_i)$.

After sampling descriptions from the on-policy LVLM, given an off-policy reasoning language model $\bar{\pi}$, as well as a query $\boldsymbol{x}_{q_i}$ and its corresponding descriptions $\boldsymbol{c}_i$, we can leverage prompt engineering to construct $\hat{\boldsymbol{x}}_{q_i}$ based on the query and description. This enables the reasoning model to simulate a LVLM that can see the image based on the provided description (see Figure A3 in Appendix E for details). As a result, we can sample the slow-thinking reasoning trajectory $\boldsymbol{y}_i \sim \bar{\pi}(\cdot|\hat{\boldsymbol{x}}_{q_i})$.

We assume that the generated $\boldsymbol{c}_i$ can effectively encode the visual cues required for reasoning. Prior studies show that, in LLaVA-style LVLMs, adapter layers project visual embeddings into text-embedding subspaces, aligning continuous visual features with their textual representations [57]. Hence, visual understanding (whether expressed as embeddings or tokens) serves as the conditioning context for autoregressive reasoning. When this context captures spatial layouts, object relations and fine-grained details accurately, the reasoning becomes more coherent and logically consistent.

## 4.2 Reward Evaluation and Propagation

**Reward for Slow Thinking.** When LLM has generated tokens $\boldsymbol{y}_{1:l}$, the policy model is in state $s_l \in \mathcal{S}$, where $\mathcal{S}$ is the finite state space and $l = 0, 1, 2, \cdots, L_y$. Since fine-grained supervision over intermediate reasoning steps is typically unavailable, a practical solution is to extract the final answer [4, 58] and evaluate its accuracy using rules or models, assigning an outcome-based reward at the end. We adopt this outcome-based reward strategy. Thus, the episodic reward for a reasoning trajectory $\boldsymbol{y}$ is defined as Eq. (4), where $r(\cdot)$ calculates the reward for each state.

$$R(\boldsymbol{y}) = \sum_{l=0}^{L_y} r(s_l), \quad r(s_l) = \begin{cases} 1 & \text{if } l = L_y \text{ and the answer is correct} \\ 0 & \text{otherwise.} \end{cases}. \tag{4}$$

**Reward for Visual Understanding.** To propagate the correctness signal to the related visual descriptions, we define the reward of each description $R(\boldsymbol{c}_i)$ based on the average outcome score of the reasoning trajectories. Specifically, we sample $N$ reasoning trajectories from the semi-off-policy behavior model $\bar{\pi}(\cdot|\hat{\boldsymbol{x}}_{q_i})$ conditioned on the visual input, and compute

$$R(\boldsymbol{c}_i) = \frac{1}{N} \sum_{j=1}^{N} R(\boldsymbol{y}_i^{(j)}). \tag{5}$$

**Reward for Off-Policy Dataset.** After obtaining both reasoning and visual reward, along with a number of reasoning trajectories regarding the image and query, we use them to construct the off-policy dataset $\mathcal{D}$, and use their rewards to guide the policy updating. The reward is defined as

$$R(\boldsymbol{x}_{q_i}, \boldsymbol{v}_i, \boldsymbol{y}_i) = \begin{cases} 1 & \text{if } R(\boldsymbol{c}_i) > \alpha, R(\boldsymbol{y}_i) = 1, \boldsymbol{y}_i = \underset{L_y}{\arg\min}\{\boldsymbol{y}|(\boldsymbol{x}_{q_i}, \boldsymbol{v}, \boldsymbol{y}) \sim \mathcal{D}, R(\boldsymbol{x}_{q_i}, \boldsymbol{v}, \boldsymbol{y}) = 1\} \\ 0 & \text{otherwise} \end{cases}, \tag{6}$$

where $\alpha \in (0, 1)$ is the threshold for the visual reward. Eq. (6) ensures that the chosen reasoning trajectory contains both sufficient correct visual information and a correct final answer. At the same time, we aim to sample the shortest possible trajectory since there exists issue of overthinking in these reasoning language models [59], and opting for the shortest response not only help models to learn a more concise reasoning structure but also mitigate redundant repetitions in the trajectory.

## 4.3 Policy Updating

We adopt off-policy optimization and the object function is Eq. (3). By differentiating the Eq. 3 with respect to $\theta$, we obtain the policy gradient Eq. (7) for optimization.

$$\begin{aligned} \nabla_\theta J(\theta) &= \nabla_\theta \mathbb{E}_{(\boldsymbol{x}, \boldsymbol{v}, \boldsymbol{y}) \sim \mathcal{D}} \left[ \frac{\pi_\theta(\boldsymbol{y}|\boldsymbol{x}, \boldsymbol{v})}{\mu(\boldsymbol{y}|\boldsymbol{x}, \boldsymbol{v})} R(\boldsymbol{x}, \boldsymbol{v}, \boldsymbol{y}) \right] \\ &= \mathbb{E}_{(\boldsymbol{x}, \boldsymbol{v}, \boldsymbol{y}) \sim \mathcal{D}} \left[ \nabla_\theta \left( \frac{\pi_\theta(\boldsymbol{y}|\boldsymbol{x}, \boldsymbol{v})}{\mu(\boldsymbol{y}|\boldsymbol{x}, \boldsymbol{v})} R(\boldsymbol{x}, \boldsymbol{v}, \boldsymbol{y}) \right) \right] \\ &= \mathbb{E}_{(\boldsymbol{x}, \boldsymbol{v}, \boldsymbol{y}) \sim \mathcal{D}} \left[ \frac{R(\boldsymbol{x}, \boldsymbol{v}, \boldsymbol{y})}{\mu(\boldsymbol{y}|\boldsymbol{x}, \boldsymbol{v})} \nabla_\theta \pi_\theta(\boldsymbol{y}|\boldsymbol{x}, \boldsymbol{v}) \right] \\ &\underset{\text{log-derivative trick}}{=\!=\!=\!=\!=\!=} \mathbb{E}_{(\boldsymbol{x}, \boldsymbol{v}, \boldsymbol{y}) \sim \mathcal{D}} \left[ \frac{\pi_\theta(\boldsymbol{y}|\boldsymbol{x}, \boldsymbol{v})}{\mu(\boldsymbol{y}|\boldsymbol{x}, \boldsymbol{v})} R(\boldsymbol{x}, \boldsymbol{v}, \boldsymbol{y}) \nabla_\theta \log \pi_\theta(\boldsymbol{y}|\boldsymbol{x}, \boldsymbol{v}) \right] \\ &\underset{\text{Eq (1)}}{=\!=\!=\!=} \mathbb{E}_{(\boldsymbol{x}, \boldsymbol{v}, \boldsymbol{y}) \sim \mathcal{D}} \left[ \frac{\pi_\theta(\boldsymbol{y}|\boldsymbol{x}, \boldsymbol{v})}{\mu(\boldsymbol{y}|\boldsymbol{x}, \boldsymbol{v})} R(\boldsymbol{x}, \boldsymbol{v}, \boldsymbol{y}) \sum_{l=1}^{L_y} \nabla_\theta \log \pi_\theta(y_l|\boldsymbol{y}_{1:l-1}, \boldsymbol{x}, \boldsymbol{v}) \right] \\ &\underset{\text{Lemma A.3}}{=\!=\!=\!=\!=} \mathbb{E}_{(\boldsymbol{x}, \boldsymbol{v}, \boldsymbol{y}) \sim \mathcal{D}} \left[ R(\boldsymbol{x}, \boldsymbol{v}, \boldsymbol{y}) \sum_{l=1}^{L_y} \nabla_\theta \log \pi_\theta(y_l|\boldsymbol{y}_{1:l-1}, \boldsymbol{x}, \boldsymbol{v}) \right] \end{aligned} \tag{7}$$

Note that off-policy RL results in a distribution mismatch between the behavior policy $\mu$ and the current LVLM policy $\pi_\theta$. Standard off-policy RL corrects this mismatch and reduces bias via importance sampling (IS). In the last step, we apply Lemma A.3, which ensures directly approximating the off-policy objective using the average reward introduces only a small and controllable bias. Detailed proof is provided in Appendix A.

# 5 Implementation

## 5.1 Policy Initialization

We utilize InternVL2.5 and InternVL3.0 in 8B and 38B scale as base models, all of which is instructed. Initially, we fine-tune the base models using general visual question-answering data, covering image description and caption. These warmed-up models then serve as the initialization for the policy model $\pi_\theta$ in SOPHIA framework. We also explore directly initializing policy models with base models without warming up and discuss the influence of different initial policy models in Appendix D. The warm-up training data consists of the open-source RLAIF-V [60] and WildVision [61] datasets. For RLAIF-V, we select the accepted samples, and for WildVision, we choose the best response. For the reasoning language model, the policy model $\pi$ is initialized by QwQ-32B-Preview [33] for easy questions and DeepSeek-R1 [17] for difficult ones.

## 5.2 Reinforcement Learning

**Data Preparation.** The visual question-answering (VQA) training dataset $\mathcal{D}_{\text{train}}$ is private and consists of around 80K questions from higher educational mathematics cources in both Chinese and English, as well as vocational education subjects, covering fields like finance and management. We also analyze SOPHIA on open-source training dataset MathV360K [7] in Section 6.4. To prevent data leakage, we filtered out any queries overlapping with the benchmark using keyword matching and LLMs. Additionally, our policy updates are mainly guided by visual reasoning trajectories, with supplementary text trajectories from Numina, MATH, and past AMC/AIME (excluding AIME 2024), and general VQA data from RLAIF-V and WildVision.

**Reward Signal.** Since reasoning language models typically produce well-structured responses, we directly extract answers using regular expressions and evaluate their correctness using the rule-based verifier from OlympiadBench [35], providing binary rewards as defined in Eq. (4).

**Algorithm and Settings.** The policy gradient for optimization is described in Eq. (7), and we omit the KL regularization term during optimization, following recent practices for instruction-tuned models to improve response length [20]. Detailed algorithms, hyperparameter configurations and prompt design strategies used during sampling and policy updates are provided in Appendix B.

# 6 Experiment

## 6.1 Evaluation Setup

**Baseline.** We evaluate multiple baselines on benchmarks. When evaluating the SOPHIA on InternVL2.5-38B and InternVL3.0-38B [40], the baseline models include closed-source models such as GPT-4.1 [37] and Claude-3.7-Sonnet [39], as well as open-source models like InternVL2.5-38B-MPO, InternVL3.0-38B-MPO [46], InternVL3.0-38B-GRPO, MM-Eureka-Qwen-32B [20], Qwen2.5-VL-72B [36], QvQ-72B-preview [62] and Virgo-72B [63]. When evaluating 8B scale, the baseline models include open-source models like InternVL2.5-8B-MPO, InternVL-3.0-8B-MPO [46], Mulberry-8B [14], MiniCPM-o-8B [60], MM-Eureka-8B [20] and Qwen2.5-VL-7B [36].

**Evaluation.** To comprehensively evaluate the performance, our benchmarks include MMMU, MMMU Pro [64], MathVista [65], MathVerse [66], DynaMath [67], MathVision[34],MV-MATH [68] and OlympiadBench [35]. We use pass@1 accuracy as the metric for evaluation under the zero-shot setting. Detailed evaluation setup is in Appendix C.

## 6.2 Main Results

**Overall Comparison.** Table 1 presents the performance of SOPHIA on InternVL2.5-38B, InternVL3.0-38B, and both open-source and closed-source models across eight benchmarks. Focusing on *Base Model and Fine-tuned Ones*, SOPHIA achieves substantial gains over the base models, with average improvements of 10.94% on InternVL2.5-38B and 8.50% on InternVL3.0-38B. On challenging benchmarks such as MathVision and OlympiadBench, it boosts performance by 26.16% and 24.45%, respectively, highlighting its strong problem-solving abilities. Furthermore, SOPHIA surpasses the mixed preference optimization (MPO) method, which is trained on 3 million examples,

| Model | College-level Question | | Math-related Question | | | Challenging Scientific Reasoning | | | Average |
|---|---|---|---|---|---|---|---|---|---|
| | MMMU | MMMU Pro | MathVista | MathVerse | DynaMath | MathVision | MV-MATH | OlympiadBench | |
| *Closed-source Model* | | | | | | | | | |
| GPT-4.1 | 73.78 | 57.57 | 70.20 | 49.49 | 72.06 | 47.53 | 39.52 | 30.38 | 55.07 |
| Cluade-3.7-Sonnet | 70.44 | 56.01 | 69.30 | 47.46 | 70.04 | 42.43 | 36.24 | 36.25 | 53.52 |
| *Representative Open-source Model* | | | | | | | | | |
| Qwen2.5-VL-72B | 66.44 | 49.65 | 74.20 | **49.37** | 67.13 | 38.29 | **38.62** | 30.24 | 51.74 |
| QvQ-72B-preview | 65.89 | 41.50 | 71.40 | 43.53 | 65.81 | 37.86 | 35.04 | 33.79 | 49.35 |
| Virgo-72B | 63.67 | 40.98 | 71.10 | 45.01 | 63.09 | 38.19 | 34.30 | 35.86 | 49.03 |
| MM-Eureka-Qwen-32B | 64.00 | 47.51 | 72.90 | 47.37 | 66.41 | 39.24 | 35.29 | 34.42 | 50.89 |
| *Base Model and Fine-tuned Ones* | | | | | | | | | |
| InternVL2.5-38B | 61.22 | 42.14 | 69.50 | 37.44 | 52.95 | 31.32 | 27.08 | 23.06 | 43.09 |
| +MPO | 63.11 | 46.41 | 74.80 | 45.18 | 60.50 | 35.56 | 29.72 | 26.62 | 47.74 |
| +SOPHIA | 66.33 | 49.65 | 74.70 | 45.30 | 63.13 | 47.96 | 35.94 | 49.22 | 54.03 |
| InternVL3.0-38B | **69.33** | 46.47 | 72.20 | 42.13 | 60.72 | 32.93 | 26.78 | 25.50 | 47.01 |
| +MPO | **69.33** | 49.08 | 74.60 | 45.18 | 65.65 | 35.39 | 27.87 | 29.01 | 49.51 |
| +GRPO | 66.67 | 49.88 | 75.00 | 43.02 | **68.22** | 42.23 | 29.82 | 32.96 | 50.98 |
| +SOPHIA | 69.00 | **52.60** | **75.20** | 47.46 | 65.33 | **49.08** | 35.44 | **49.95** | **55.51** |
| +SOPHIA+GRPO | 70.22 | 56.59 | 76.60 | 49.11 | 65.87 | 50.40 | 37.63 | 51.79 | 57.28 |

Table 1: Pass@1 accuracy of SOPHIA in 38B scale, other fine-tuning strategies and baselines. In each column, An entry is in bold if its value is the best among open-source models, and underlined if it ranks second (exclude the last line "SOPHIA+GRPO").

| Model | College-level Question | | Math-related Question | | | Challenging Scientific Reasoning | | | Average |
|---|---|---|---|---|---|---|---|---|---|
| | MMMU | MMMU Pro | MathVista | MathVerse | DynaMath | MathVision | MV-MATH | OlympiadBench | |
| Mulberry-8B | 41.56 | 19.31 | 53.70 | 17.89 | 32.40 | 20.59 | 10.65 | 3.27 | 24.92 |
| MiniCPM-o-8B | 45.44 | 29.64 | 66.90 | 27.79 | 42.38 | 18.19 | 22.30 | 8.22 | 32.61 |
| MM-Eureka-8B | 51.22 | 33.35 | 67.40 | 29.82 | 39.46 | 21.97 | 20.76 | 14.97 | 34.87 |
| Qwen2.5-VL-7B | 52.00 | **38.61** | 68.90 | 38.58 | 54.35 | 26.38 | **28.42** | 16.28 | 40.44 |
| InternVL2.5-8B | 48.22 | 28.61 | 64.30 | 26.78 | 39.14 | 21.02 | 20.61 | 12.90 | 32.70 |
| +MPO | 54.33 | 32.54 | 69.40 | 29.82 | 46.07 | 22.11 | 23.05 | 14.25 | 36.45 |
| +SOPHIA | 57.00 | 32.02 | 61.10 | 30.84 | 46.51 | 22.66 | 22.00 | 18.53 | 36.33 |
| InternVL3.0-8B | 51.11 | 33.99 | 67.80 | 33.12 | 47.34 | 25.66 | 21.95 | 19.40 | 37.55 |
| +MPO | 58.33 | 37.69 | **72.60** | **39.72** | **56.77** | 35.40 | 26.18 | 20.63 | 43.41 |
| +SOPHIA | **60.67** | 38.27 | 64.40 | 35.91 | 55.39 | **37.89** | 26.48 | **38.67** | **44.71** |

Table 2: Pass@1 accuracy of SOPHIA in 8B scale and baselines in small scale. In each column, An entry is in bold if its value is the best, and underlined if it ranks second.

showing more efficient data utilization and greater effectiveness than pipeline-based supervised fine-tuning. Overall, InternVL3.0-38B + SOPHIA achieves the best or second-best pass@1 accuracy on most benchmarks and demonstrates state-of-the-art performance in terms of the average of these benchmarks, coming close to the performance of the closed-source model GPT 4.1. Notably, our model achieves a remarkable pass@1 accuracy of 49.08 on MathVision and 49.95 on OlympiadBench.

**On-Policy RL.** On-policy RL is commonly used to enhance slow-thinking reasoning. The policy initialization includes both the original InternVL3.0-38B and the model trained using SOPHIA. The training queries are drawn from a subset of the VQA training dataset used in SOPHIA. Detailed GRPO configurations are in Appendix B.2. As shown in Figure 3, we track pass@1 accuracy for InternVL3.0-38B and InternVL3.0-38B+SOPHIA on MMMU Pro and MathVision over 480 training steps. The solid lines show each model's performance; dashed lines indicate base model results from Table 1. While performance of InternVL3.0-38B declines with GRPO, SOPHIA not only prevents this drop but also improves accuracy. This highlights the limitations of on-policy RL for LVLMs (Section 1) and shows that our method enables stronger policy initialization for further on-policy RL.

Moreover, we evaluate the checkpoint with the highest average score on MathVision and MMMU Pro after 100 steps. As shown in Table 1, SOPHIA outperforms GRPO across multiple benchmarks, showing stronger visual slow-thinking reasoning. Further on-policy RL boosts performance beyond direct RL, achieving Claude-3.7-Sonnet on college-level tasks and surpassing most closed-source models on challenging scientific reasoning.

**Overall Performance in Small Scale.** The performance of SOPHIA on 8B scale and other open-source models are shown in Table 2. At the 8B scale, SOPHIA outperforms the base model across most benchmarks. It significantly surpasses the MCTS-based Mulberry 8B and remains competitive with both the GRPO-based MM-Eureka 8B and MPO-based methods. Notably, InternVL 3.0 8B+SOPHIA achieves the best on average performance and excels on college-level MMMU and challenging OlympiadBench. In contrast, SOPHIA performs worse on MathVista. We attribute this to

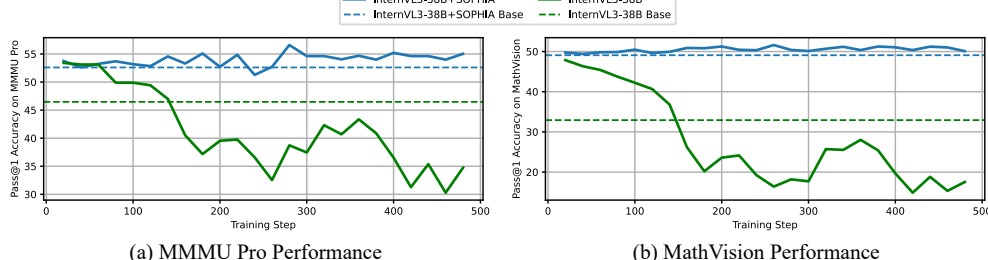

|  | (a) MMMU Pro Performance | (b) MathVision Performance |
|---|---|---|

Figure 3: The performance of InternVL3.0-38B and InternVL3.0-38B+SOPHIA on MMMU Pro and MathVision over 480 training steps, where the green and blue solid lines represent their respective pass@1 accuracy, while the dashed lines indicate the base model performance.

| Model | DynaMath | | MathVision | | OlympiadBench | |
|---|---|---|---|---|---|---|
| | Plane Geo | Solid Geo | Plane Geo | Solid Geo | Plane Geo | Solid Geo |
| GPT-4.1 | 59.87 | 54.00 | 50.75 | 43.08 | 33.08 | 13.65 |
| QvQ-72B-preview | 58.05 | 55.33 | 40.77 | 30.74 | 33.49 | 24.32 |
| InternVL2.5-38B | 43.90 | 25.33 | 33.99 | 26.23 | 12.71 | 4.16 |
| +SOPHIA | 62.08 | 67.33 | 56.95 | **45.49** | 50.55 | 52.54 |
| InternVL3.0-38B | 51.43 | 46.67 | 36.92 | 30.74 | 22.64 | 12.61 |
| +SOPHIA | **62.99** | **72.00** | **58.26** | 43.85 | **56.25** | **62.81** |

Table 3: Pass@1 accuracy of SOPHIA and baselines in geometry questions. "Plane Geo" in MathVision includes metric geometry-angle, metric geometry-area and metric geometry-length. "Plane Geo" in OlympiadBench includes geometry and plane geometry. In each column, An entry is in bold if its value is the best, and underlined if it ranks second.

| Method | MMMU | MathVista | MathVision | OlympiadBench | Average |
|---|---|---|---|---|---|
| InternVL2.5-38B + SOPHIA | | | | | |
| freeze ViT | **66.33** | **74.70** | 47.96 | 49.22 | **59.55** |
| unfreeze ViT | 65.44 | 72.40 | **48.68** | **49.53** | 59.01 |
| InternVL3.0-38B + SOPHIA | | | | | |
| freeze ViT | **69.00** | **75.20** | 49.08 | 49.95 | **60.81** |
| unfreeze ViT | 67.67 | 71.60 | **49.34** | **50.51** | 59.78 |

Table 4: Ablation study on ViT freezing when using SOPHIA. A bold entry indicate better performance compared to the alternative under the same base model.

| Method | MMMU | MathVista | MathVision | OlympiadBench | Average |
|---|---|---|---|---|---|
| InternVL2.5-38B | 61.22 | 69.50 | 31.32 | 23.06 | 46.28 |
| +Random | 64.00 | 69.70 | 37.32 | 35.10 | 51.53 |
| +SOPHIA w/o CR | 64.11 | 69.80 | 40.35 | 39.89 | 53.54 |
| +SOPHIA w/o S | **66.33** | 70.10 | 41.65 | 40.86 | 54.74 |
| +SOPHIA | 65.44 | **70.90** | **42.73** | **41.46** | **55.13** |

Table 5: Ablation study on reward design. "Random" randomly samples trajectories from $\mathcal{D}$; "SOPHIA w/o CR" removes the caption reward; "SOPHIA w/o S" drops the shortest trajectory constraint. Best value is in bold.

the training data's limited coverage of certain problem types, such as age-difference reasoning. Due to the smaller constrained generalization capability of model, it struggles to develop robust reasoning strategies for these underrepresented cases, resulting in lower performance.

**Geometry Performance.** Geometry is a challenging math tasks and requires both visual understanding and reasoning [69]. We evaluate SOPHIA in 38B scalre as well as top models and the base model on DynaMath, MathVision, and OlympiadBench geometry problems. As shown in Table 3, SOPHIA outperforms other open-source models, especially on complex OlympiadBench tasks, validating its effectiveness in enhancing the abilities of leveraging the visual understanding during reasoning.

## 6.3 Ablation Study

We assess the impact of weight freezing, and the effectiveness of our proposed reward and warm-up. Further experiments on response style, teacher models, and caption as training data are detailed in Appendix D.

**Frozen/Unfrozen ViT.** We conduct experiments on InternVL2.5-38B and InternVL3.0-38B to explore the effect of freezing the visual transformer. As shown in Table 4, freezing the visual transformer consistently improves average performance, with notable gains on general-purpose benchmarks such as MMMU. We attribute this to the possibility that long reasoning trajectories may interfere with

| Method | College-level Question | | Math-related Question | | | Challenging Scientific Reasoning | | | Average |
|---|---|---|---|---|---|---|---|---|---|
| | MMMU | MMMU Pro | MathVista | MathVerse | DynaMath | MathVision | MV-MATH | OlympiadBench | |
| InternVL2.5-38B | 61.22 | 42.14 | 69.50 | 37.44 | 52.95 | 31.32 | 27.08 | 23.06 | 43.09 |
| +SOPHIA w/o warm-up | 65.66 | 48.96 | 70.90 | **45.56** | 61.08 | 44.21 | 34.60 | 44.10 | 51.88 |
| +SOPHIA | **66.33** | **49.65** | **74.70** | 45.30 | **63.13** | **47.96** | **35.94** | **49.22** | **54.03** |
| InternVL2.5-8B | 48.22 | 28.61 | **64.30** | 26.78 | 39.14 | 21.02 | 20.61 | 12.90 | 32.70 |
| +SOPHIA w/o warm-up | 51.44 | 30.98 | 60.10 | **30.84** | 45.35 | **23.75** | 17.47 | 17.54 | 34.68 |
| +SOPHIA | **57.00** | **32.02** | 61.10 | **30.84** | **46.51** | 22.66 | **22.00** | **18.53** | **36.33** |

Table 6: Ablation study of warm-up on InternVL2.5-38B and InternVL2.5-8B. A bold entry indicate best performance compared to others under the same base model.

vision-language alignment. Therefore, we freeze the visual transformer during training to not only improve performance but also enhance training efficiency.

**Reward.** In Eq.6, the reward includes not only for the correctness of the reasoning trajectory but also for the quality of visual information. To evaluate the effectiveness, we conduct ablation studies on InternVL2.5-38B. To ensure the validity of our results, no additional general visual or textual data are used. As shown in Table 5, "Random" refers to trajectories randomly sampled from off-policy dataset $\mathcal{D}$; "SOPHIA w/o CR" excludes the caption reward and uses only the outcome reward; "SOPHIA w/o S" removes the constraint that we select shortest one. We observe that: **1)** The Random strategy is a strong baseline; its modest gains on MathVision and OlympiadBench suggest that even imperfect reasoning can help models learn reasoning patterns. **2)** Using outcome reward promotes correct logic, but unfiltered visual errors still limit performance on difficult datasets. **3)** Selecting the shortest trajectories helps avoid repetition and redundancy, yielding notable gains on challenging datasets.

**Warm up.** To investigate the impact of the warm-up on the performance, we set up an experimental group including "SOPHIA w/o warm-up" and "SOPHIA w/o warm-up". The policy initialization in the experimental group is directly InternVL2.5-38B (Instruct) and InternVL2.5-8B (Instruct) without a warm-up phase. The control group includes models trained following the standard procedure. The experimental results are shown in Table 6. We observe that models with warm-up consistently outperform those without warm-up. This is likely due to the emphasis of warm-up data on visual description and grounding. Such training enhances the captioning ability of model, thereby improving the quality of trajectories collected by the behavior model.

## 6.4 Analysis of Training Data

We further evaluate SOPHIA on different training datasets and data scales. Additional experiments including caption data mixing, different data sources, and the Keep-$N$ strategy, are in Appendix D.

**Versatility.** To verify that performance improvements are not solely due to higher data quality, we also evaluate our method on the public

| Method | MMMU | MathVista | MathVision | OlympiadBench | Average |
|---|---|---|---|---|---|
| InternVL2.5-38B | 61.22 | 69.50 | 31.32 | 23.06 | 46.28 |
| + SOPHIA | | | | | |
| MathV360K 10% | 64.89 | 67.60 | 42.01 | 39.17 | 53.42 |
| MathV360K 50% | **67.33** | 66.90 | 42.14 | 40.18 | 54.14 |
| Private | 65.44 | **70.90** | **42.73** | **41.46** | **55.13** |

Table 7: Versatility analysis of SOPHIA.

MathV360K dataset. Specifically, using the same settings as with the private dataset (80K), we train InternVL2.5-38B on 10% (36K) and 50% (180K) subsets of MathV360K to explore the trade-off between data quality and quantity. No additional general visual or textual data are used to ensure a fair comparison. Results are shown in Table 7. We observe that SOPHIA still brings performance gains when trained on open-source datasets. However, despite containing more questions than private data, these datasets do not lead to notable improvements on MathVision and OlympiadBench, even degrading on MathVista. This suggests that the quality and difficulty of questions are more critical than the quantity.

**Scaling Law.** We examine how SOPHIA scales with data size by training InternVL2.5-38B on datasets of 5K, 10K, 20K, 40K, and 80K questions. To ensure a fair comparison, no additional general-purpose visual or textual data are used. The results are shown in Table 8. We observe that: **1)** Average performance improves with more data, showing rapid gains initially and diminishing returns thereafter; **2)** Larger datasets yield significant improvements on more challenging benchmarks; **3)** For MMMU and MMMU Pro, performance is sensitive to data size only at smaller scales, with limited gains beyond that; **4)** On MathVista, MathVerse and DynaMath, increasing the dataset size does not consistently lead to performance gains and may even be detrimental in some cases.

| #Data | College-level Question | | Math-related Question | | | Challenging Scientific Reasoning | | | Average |
|---|---|---|---|---|---|---|---|---|---|
| | MMMU | MMMU Pro | MathVista | MathVerse | DynaMath | MathVision | MV-MATH | OlympiadBench | |
| 5K | 59.22 | 44.22 | 67.20 | 39.09 | 60.69 | 35.26 | 28.42 | 28.80 | 45.36 |
| 10K | 65.22 | 46.47 | 70.10 | 38.83 | 60.35 | 37.50 | 30.66 | 34.80 | 47.99 |
| 20K | 66.56 | 45.03 | 71.80 | 44.80 | 60.82 | 41.41 | 31.61 | 38.80 | 50.10 |
| 40K | 66.11 | 47.57 | 70.40 | 46.45 | 58.40 | 41.41 | 32.45 | 39.59 | 50.30 |
| 80K | 65.44 | 46.93 | 70.90 | 45.18 | 59.14 | 42.73 | 34.15 | 41.46 | 50.74 |

Table 8: Pass@1 accuracy of SOPHIA on InternVL 2.5 38B in different data scale.

## 6.5 Qualitative Analysis on Hallucinations and Logical Inconsistencies

We conduct manual evaluation on 50 randomly sampled cases from MMMU Pro before and after SOPHIA training, covering diverse disciplines to assess both visual hallucination and logical inconsistency. We report the proportion of responses with visual hallucinations and logic inconsistency. As

| Method | Visual Hallucination | Logic Inconsistency |
|---|---|---|
| InternVL3.0-38B | 38% | 54% |
| InternVL3.0-38B+SOPHIA | 16% | 44% |

Table 9: Qualitative analysis of SOPHIA.

shown in Table 9, SOPHIA significantly reduces hallucinations and logic error, consistent with improvements in overall correctness, which supports our motivation that on-policy visual understanding training can enhance the visual understanding of the model while reducing hallucinations.

## 7 Conclusion and Future Work

This paper presents SOPHIA, a simple and scalable semi-off-policy reinforcement learning framework to enhance the visual slow-thinking reasoning capabilities of large vision-language models (LVLMs). By leveraging a semi-off-policy behavior model that integrates on-policy visual understanding with off-policy slow-thinking reasoning, SOPHIA not only enhances the LVLM's slow-thinking abilities but also uses visual rewards to guide the LVLM in optimizing the strategy of leveraging its own visual understanding during reasoning without requiring closed-source or human-annotated data. Extensive experiments across eight multimodal benchmarks show that SOPHIA achieves state-of-the-art performance among open-source LVLMs and even surpasses some closed-source models on challenging benchmarks like MathVision and OlympiadBench. SOPHIA also outperforms both supervised fine-tuning and on-policy RL methods, serving as a better initialization for further on-policy reinforcement learning, enabling continual improvement without performance degradation.

Despite these advances, SOPHIA still struggles with long-range visual dependencies and fine-grained or low-level recognition in complex scenes. Also, performance may suffer when facing underrepresented domains or noisy samples. Issues such as hallucinated visual content, redundant reasoning, and the binary nature of reward signals still persist. Future work will explore stronger visual encoders, adaptive curriculum learning, and more robust data augmentation to further improve generalization, efficiency, and reasoning robustness across broader domains.

## Acknowledgements

We would like to thank the anonymous reviewers and area chairs for their comprehensive and constructive reviews. This work is supported by the Shanghai Artificial Intelligence Laboratory. The authors would like to thank Yiming Zhang, Chengqi Lyu and Jiangning Liu to their valuable suggestions and comments on this work.

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

## A    Theoretical Analysis

We first explore the properties of trajectories in off-policy dataset $\mathcal{D}$.

**Lemma A.1.** *There exits a set $\mathcal{Y}$ such that $\forall \boldsymbol{y} \notin \mathcal{Y}$, either $\mu(\boldsymbol{y}|\hat{\boldsymbol{x}}_q, \boldsymbol{v}) = 0$ or $R(\boldsymbol{x}_q, \boldsymbol{v}, \boldsymbol{y}) = 0$.*

*Proof.* Let the set $\mathcal{Y}$ be defined as

$$\mathcal{Y} = \{\boldsymbol{y}|(\boldsymbol{x}_q, \boldsymbol{v}, \boldsymbol{y}) \sim \mathcal{D}, R(\boldsymbol{x}_q, \boldsymbol{v}, \boldsymbol{y}) = 1\}. \tag{A1}$$

We aim to show that for any trajectory $\boldsymbol{y} \notin \mathcal{Y}$ and any context $(\hat{\boldsymbol{x}}_q, \boldsymbol{v}, \boldsymbol{x}_q)$, either $\bar{\pi}(\boldsymbol{y}|\hat{\boldsymbol{x}}_q) = 0$ or $R(\boldsymbol{x}_q, \boldsymbol{v}, \boldsymbol{y}) = 0$. Consider an arbitrary trajectory $\boldsymbol{y}$ such that $\boldsymbol{y} \notin \mathcal{Y}$, and an arbitrary context $(\hat{\boldsymbol{x}}_q, \boldsymbol{v}, \boldsymbol{x}_q)$. If $\bar{\pi}(\boldsymbol{y}|\hat{\boldsymbol{x}}_q) = 0$, the condition of lemma is immediately satisfied.

Alternatively, assume $\bar{\pi}(\boldsymbol{y}|\hat{\boldsymbol{x}}_q) > 0$. We then need to demonstrate that $R(\boldsymbol{x}_q, \boldsymbol{v}, \boldsymbol{y}) = 0$. If the specific tuple $(\boldsymbol{x}_q, \boldsymbol{v}, \boldsymbol{y})$ (for which we assume $\bar{\pi}(\boldsymbol{y}|\hat{\boldsymbol{x}}_q) > 0$) is indeed a sample in $\mathcal{D}$. Then, because $\boldsymbol{y} \notin \mathcal{Y}$, by the definition of $\mathcal{Y}$, this sample $(\boldsymbol{x}_q, \boldsymbol{v}, \boldsymbol{y})$ cannot satisfy $R(\boldsymbol{x}_q, \boldsymbol{v}, \boldsymbol{y}) = 1$. If it did, $\boldsymbol{y}$ would be in $\mathcal{Y}$, which contradicts our premise. Since the reward $R(\cdot)$ can only be 0 or 1, it follows that $R(\boldsymbol{x}_q, \boldsymbol{v}, \boldsymbol{y}) = 0$.

This completes the proof.    $\square$

The proof gives a special subset of all trajectories in $\mathcal{D}$, *i.e.*, $\mathcal{Y} = \{\boldsymbol{y}|(\boldsymbol{x}_q, \boldsymbol{v}, \boldsymbol{y}) \sim \mathcal{D}, R(\boldsymbol{x}_q, \boldsymbol{v}, \boldsymbol{y}) = 1\}$, whose property results from our appropriate sampling strategy and reward design. Thus, Assumption A.2 holds. This assumption ensures the coverage ($\mu(\boldsymbol{y}|\hat{\boldsymbol{x}}_q, \boldsymbol{v}) > 0$) and that $\pi_\theta$ and $\mu$ do not deviate significantly from each other within $\mathcal{Y}$.

**Assumption A.2.** $\forall \boldsymbol{y} \in \mathcal{Y}, \mu(\boldsymbol{y}|\hat{\boldsymbol{x}}_q, \boldsymbol{v}) > 0$, and the following boundedness condition holds

$$\left|\frac{\pi_\theta(\boldsymbol{y}|\boldsymbol{x}_q, \boldsymbol{v})}{\mu(\boldsymbol{y}|\boldsymbol{x}_q, \boldsymbol{v})} - 1\right| \leq \delta, \tag{A2}$$

where $\delta > 0$ is a small constant.

Finally, we present Lemma A.3 to address aforementioned issues.

**Lemma A.3.** *Given the expectation using IS and without IS (i.e., treating the ratio as 1)*

$$G_{IS} = \mathbb{E}_{(\boldsymbol{x}_q, \boldsymbol{v}, \boldsymbol{y}) \sim \mathcal{D}}\left[\frac{\pi_\theta(\boldsymbol{y}|\boldsymbol{x}_q, \boldsymbol{v})}{\mu(\boldsymbol{y}|\boldsymbol{x}_q, \boldsymbol{v})}R(\boldsymbol{x}_q, \boldsymbol{v}, \boldsymbol{y})\right], \quad G_I = \mathbb{E}_{(\boldsymbol{x}_q, \boldsymbol{v}, \boldsymbol{y}) \sim \mathcal{D}}\left[R(\boldsymbol{x}_q, \boldsymbol{v}, \boldsymbol{y})\right], \tag{A3}$$

*the bias from ignoring IS is bounded by $|G_{IS} - G_1| \leq \delta$.*

*Proof.* The difference between $G_{\text{IS}}$ and $G_1$ is

$$\begin{aligned}|G_{\text{IS}} - G_1| &= \left|\mathbb{E}_{(\boldsymbol{x}_q, \boldsymbol{v}, \boldsymbol{y}) \sim \mathcal{D}}\left[\left(\frac{\pi_\theta(\boldsymbol{y}|\boldsymbol{x}_q, \boldsymbol{v})}{\mu(\boldsymbol{y}|\boldsymbol{x}_q, \boldsymbol{v})} - 1\right)R(\boldsymbol{x}_q, \boldsymbol{v}, \boldsymbol{y})\right]\right| \\ &\leq \mathbb{E}_{(\boldsymbol{x}_q, \boldsymbol{v}, \boldsymbol{y}) \sim \mathcal{D}}\left[\left|\frac{\pi_\theta(\boldsymbol{y}|\boldsymbol{x}_q, \boldsymbol{v})}{\mu(\boldsymbol{y}|\boldsymbol{x}_q, \boldsymbol{v})} - 1\right||R(\boldsymbol{x}_q, \boldsymbol{v}, \boldsymbol{y})|\right].\end{aligned} \tag{A4}$$

Since $\boldsymbol{y} \notin \mathcal{Y}$ contributes nothing to the expectation, we only need to consider $\boldsymbol{y} \in \mathcal{Y}$. Using an indicator function $\mathbf{1}(\boldsymbol{y} \in \mathcal{Y})$, we have

$$|G_{\text{IS}} - G_1| \leq \mathbb{E}_{(\boldsymbol{x}_q, \boldsymbol{v}, \boldsymbol{y}) \sim \mathcal{D}}\left[\mathbf{1}(\boldsymbol{y} \in \mathcal{Y})\left|\frac{\pi_\theta(\boldsymbol{y}|\boldsymbol{x}_q, \boldsymbol{v})}{\mu(\boldsymbol{y}|\boldsymbol{x}_q, \boldsymbol{v})} - 1\right||R(\boldsymbol{x}_q, \boldsymbol{v}, \boldsymbol{y})|\right]. \tag{A5}$$

Applying the bounds $|R(\boldsymbol{x}_q, \boldsymbol{v}, \boldsymbol{y})| \leq 1$ and Assumption A.2, we have

$$|G_{\text{IS}} - G_1| \leq \mathbb{E}_{(\boldsymbol{x}_q, \boldsymbol{v}, \boldsymbol{y}) \sim \mathcal{D}} \left[ \mathbf{1}(\boldsymbol{y} \in \mathcal{Y}) \delta \cdot 1 \right] = \delta \mathbb{E}_{(\boldsymbol{x}_q, \boldsymbol{v}, \boldsymbol{y}) \sim \mathcal{D}} \left[ \mathbf{1}(\boldsymbol{y} \in \mathcal{Y}) \right]$$
$$\leq \delta \cdot 1 = \delta \,. \tag{A6}$$

This completes the proof. $\square$

Lemma A.3 shows that, under the above assumption, directly approximating the off-policy objective using the average reward introduces only a small and controllable bias.

# B    Implementation Details

## B.1    SOPHIA Setup

**Algorithm.** The reinforcement learning training procedure is shown in Algorithm 1. To implement the proposed, we perform a single round of policy update after collecting all trajectory samples, aiming to ensure training stability, improve computational efficiency, and mitigate excessive distributional shift when captioning images.

---

**Algorithm 1** The Semi-off-Policy Reinforcement Learning Algorithm

---
1: **Inputs:** Training dataset $\mathcal{D}_{\text{train}}$, policy model (LVLM) $\pi_\theta$, reasoning language model $\bar{\pi}$, number of rollouts per image $K$, number of rollouts per caption $N$.
2: Initialize policy $\pi_\theta$ and $\bar{\pi}$ as Section 5.1.
3: **for** $I \in \mathcal{D}_{\text{train}}$ **do**
4:     Obtain its image tokens $\boldsymbol{v}$.
5:     Generate $K$ caption samples $\{\boldsymbol{c}^{(1)}, \boldsymbol{c}^{(2)}, \cdots, \boldsymbol{c}^{(K)}\}$ where $\boldsymbol{y}^{(k)} \sim \pi_i(\cdot|\boldsymbol{x}_d, \boldsymbol{v})$.
6:     **for** $k = 1, 2, \cdots K$ **do**
7:         Construct prompt $\hat{\boldsymbol{x}}_q$ based on $\boldsymbol{c}^{(k)}, \boldsymbol{x}_q$.
8:         Sample $N$ reasoning trajectories $\{\boldsymbol{y}^{(1)}, \boldsymbol{y}^{(2)}, \cdots, \boldsymbol{y}^{(N)}\}$ where $\boldsymbol{y}^{(n)} \sim \bar{\pi}(\cdot|\hat{\boldsymbol{x}}_{q_i})$
9:         Calculate the reward of each trajectory with Eq. (4).
10:     **end for**
11:     Calculate the reward of each caption with Eq. (5).
12: **end for**
13: Construct the off-policy dataset $\mathcal{D}$ with their reward based on Eq. 6.
14: Update $\pi_\theta$ with Eq. (7).
15: **Return:** The optimized policy model $\pi^*$.

---

**KL Regularization.** Some earlier studies on RLHF [70, 71] indicate that incorporating a KL regularization term helps prevent the model from overly prioritizing reward maximization, which could otherwise compromise the linguistic patterns acquired during pretraining. However, recent research on on-policy RL-based methods shows that omitting the KL regularization term during fine-tuning an instruction-tuned model reduces computation and increases response length [20]. Considering that the policy initialization is instruction-tuned model, we follow this approach.

**Hyperparameters.** During reasoning-rewarded sampling, we set $K = N = 8$. The max length of each caption and rollout trajectory is 32678 tokens. During policy updating, the threshold of caption reward $\alpha = 0.75$, and only the weights of language backbone of the LVLM are unfrozen. The policy model is trained with batchsize of 512, learning rate of $2 \times 10^{-5}$, weight decay of 0.05 and employs a cosine annealing learning rate schedule, decaying to $1/4$ of the initial learning rate over time. We optimize the policy model using the AdamW optimizer. When training, we apply different system prompts to distinguish data from different sources (see Appendix E for more details).

## B.2    GRPO Steup

**Data Preparation.** All data used in this section are drawn from our VQA training set, without incorporating any general visual or text-only training data. Each question is independently sampled 16 times using the InternVL3.0-38B model to estimate its correctness rate. To improve the effectiveness of on-policy RL, we retain only questions with correctness rates between 0.4 and 0.6, filtering out those that are too easy or too difficult.

**Training Details.** We adopt Qwen2.5-72B-Instruct as the generative verifier and use a binary outcome reward signal for GRPO. During training, each batch contains 64 questions, with 8 rollouts per question and a maximum trajectory length of 16384 tokens. The correctness scores across rollouts are averaged to compute a pass rate; questions with a pass rate of exactly 0 or 1 are excluded, using 0.5 as the threshold for incorrectness. The policy model is trained with a learning rate of $5 \times 10^{-7}$, with all other settings consistent with Appendix B.

### B.3 Computational Costs

During the full training process, the computational cost of training the InternVL3.0-38B model serves as the baseline. Under the same data scale, the proposed GRPO-based training consumes roughly three-quarters of that baseline's total computation. For inference, the SOPHIA-trained model follows the same algorithmic procedure as the base LVLM (*i.e.*, next-token prediction). However, due to its slow-thinking and step-by-step reasoning design, it requires a longer reasoning sequence to reach the final output. Empirically, the inference process of SOPHIA is about 3-4 times slower than that of the base LVLM on standard reasoning benchmarks. This moderate increase aligns with the "test-time scaling" paradigm in reasoning models

## C Evaluation Details

**Benchmark.** To comprehensively evaluate the performance of model, our benchmarks include college-level question, math-related question and challenging scientific reasoning. The specific split of datasets is shown as below.
- MMMU [64]: the valiation split of MMMU dataset.
- MMMU Pro [64]: the vision split of MMMU Pro dataset, where the input image contains both the visual content and the question, while the text query excludes the question.
- MathVista [65]: the testmini split of MathVista dataset.
- MathVerse [66]: the testmini and vision-only split of MathVerse dataset.
- DynaMath [67]: the full test set of DynaMath dataset.
- MathVision [34]: the full test set of MathVision dataset.
- MV-MATH [68]: the full test set of MV-MATH dataset.
- OlympiadBench [35]: the full test set of OlympiadBench dataset, including visual and texture questions and excluding all proof questions.

**Metrics and Evaluation.** We use pass@1 accuracy as the metric for evaluation under the zero-shot setting. Since greedy decoding may lead to performance degradation and endless repetitions for reasoning model, models are evaluated with temperature of 1.0, top-p of 0.8 and the output token length of 16384. To ensure optimal performance for all models, a response hint is added before the original query if the benchmark provides it. For all benchmarks across all models, answers are extracted using Qwen2.5-72B-Instruct [72] with greedy decoding (see Figure A5 in Appendix E for the extraction prompt). For benchmarks where answer formats are difficult to match with rules (*i.e.*, MMMU, MathVerse and MV-MATH), answers are evaluated using Qwen2.5-72B-Instruct (see Figure A6 in Appendix E for evaluation prompt); for others, verifier from OlympiadBench is used for strict equivalence checking.

## D Additional Experiment on SOPHIA

**Response Style and Teacher Model.** For the behavior policy, we evaluate different slow-thinking reasoning styles and their impact on SOPHIA performance. As shown in Fig.A1, the QwQ Preview style presents the full reasoning process without a summary or <think>...</think> tags, while the DeepSeek-R1 style is more decisive and uses these tags to separate reasoning from summary. Experiments are conducted on InternVL2.5-38B and no additional general visual or textual data are used. Results are shown in Table A1. We observe no significant differences between two different models in terms of quality (DeepSeek-R1 > QwQ-32B-Preview) and size (658B > 32B), indicating that the focus at trajectories should be on the correctness and logical consistency of trajectories rather than style or format, as well as SOPHIA can achieve strong performance without an extremely powerful teacher model. In practice, we combine both response styles: QwQ Preview is used for simpler questions, while DeepSeek-R1 handles more complex ones.

Alright, I've got this geometry problem here. Let's see what it's asking. The question is about triangle ABC, and it mentions two angle bisectors, OB and OC, which intersect at point O. It gives me that angle A is 110 degrees, and I need to find the measure of angle BOC. The choices are 135°, 140°, 145°, or 150°.

First, I need to make sure I understand the setup. We have triangle ABC, with angles at vertices A, B, and C. OB and OC are the angle bisectors of angles B and C, respectively, and they intersect at point O inside the triangle.
Wait a minute, angle BOC is the...
Wait a minute, angle BOC is the...
Alternatively, I can consider the properties...
...
So, the answer should be 145°, which corresponds to option C.

I think that's solid. I've used two different methods to arrive at the same answer, which gives me confidence in the result.
**Final Answer** \\[ \\boxed{C} \\]

(a) QwQ Preview Style

<think>
Okay, let's tackle this problem step by step. The question is: Subtract all large rubber spheres. Subtract all big shiny cylinders. How many objects are left? Hmm, first, I need to identify all the objects in the image and categorize them into large rubber spheres, large shiny cylinders, and the rest.

Looking at the image description: there's a green large rubber sphere, a purple large rubber cylinder, a gray large metal cube, a yellow large rubber cube, a yellow large metal cylinder, and a purple small rubber sphere.
...
Wait, just to make sure I didn't miss any objects. The description lists all six, so yes. Subtracting the two mentioned objects gives 4. So the answer should be 4.
</think>
To determine how many objects are left after subtracting the large rubber sphere and the large shiny cylinder, follow these steps:...
**Answer:** 4

(b) DeepSeek-R1 Style

Figure A1: Examples of different response styles. (a) is the QwQ Preview response style and (b) is the DeepSeek-R1 response style.

**Caption as Training Data.** SOPHIA generates a large number of image-related captions during sampling. We explore whether incorporating these captions into model fine-tuning improves performance. Experiments are conducted on InternVL2.5-38B, using only captions with a reward score of $R(c) = 1.0$, without adding any additional general visual or textual data. As shown in Table A2, "SOPHIA DI" refers to training data composed of the query, image, and caption. "SOPHIA H" denotes a setting that the sampled caption is prepended to the reasoning trajectory and connected by "\n\n" to construct a new trajectory for policy optimization. We observe that incorporating captions offers no significant benefit, and placing them before reasoning trajectories can even hurt performance. This degradation stems from two factors: **1)** captions consume part of the output token budget. **2)** their distribution diverges from that of reasoning trajectories. Moreover, reasoning trajectories already embed the necessary visual understanding. Thus, caption data is not incorporated into the training dataset.

| Method | MMMU | MathVista | MathVision | OlympiadBench | Average |
|---|---|---|---|---|---|
| InternVL2.5-38B + SOPHIA | | | | | |
| QwQ Preview | 65.89 | 67.30 | 42.34 | 43.71 | 54.81 |
| DeepSeek-R1 | 64.56 | 68.50 | 43.78 | 42.63 | 54.87 |

Table A1: Ablation study of the response style for SOPHIA. "QwQ Preview" represents the QwQ Preview response style (Figure A1 (a)) and "DeepSeek-R1" is the DeepSeek-R1 response style (Figure A1 (b)).

**Data Composition.** To examine the interaction between different data sources, we incorporate general visual data (GV) and pure textual data (Text) during policy updates. We conduct experiments on InternVL2.5-38B, and the results are summarized in Table A3. In the table, "GV" and "Text" indicate the general visual and textual data, respectively, while "RC" represents the trajectory and update strategy from SOPHIA. A ✓ denotes that the corresponding data type is used in the experiment. We observe that text trajectories alone can provide moderate improvements in reasoning abilities; however, the absence of visual understanding limits their effectiveness. Introducing general visual data or SOPHIA alongside text leads to further gains, with the best results achieved when all three sources are combined.

| Method | MMMU | MathVista | MathVision | OlympiadBench | Average |
|---|---|---|---|---|---|
| InternVL2.5-38B | 61.22 | 69.50 | 31.32 | 23.06 | 46.28 |
| + SOPHIA DI | 65.11 | 69.60 | 41.18 | 39.94 | 53.26 |
| + SOPHIA H | **65.89** | 67.00 | 40.89 | 39.27 | 53.96 |
| + SOPHIA | 65.44 | **70.90** | **42.73** | **41.46** | **55.13** |

Table A2: Ablation study of caption as training data for SOPHIA. "SOPHIA DI" represents training data composed of the query, image, and caption. "SOPHIA H" represents that sampled caption is prepended to the reasoning trajectory to construct a new trajectory. Best value is in bold.

**Keep-$N$.** In our trajectory selection, we prioritize the shortest valid trajectory, which effectively implements a Keep-1 strategy[1]. A Keep-$N$ strategy selects the top-$N$ shortest trajectories for each image-query pair, as long as they are correct and their caption rewards exceed the threshold. This approach allows up to $N$ high-quality trajectories to be used for policy updates. This strategy may enhance learning in visual scenarios, and we evaluate its effectiveness on InternVL2.5-38B. Here, no additional general visual or textual data are introduced. Results are presented in Table A4. We find that the Keep-$N$ strategy can improve model performance to some extent, *e.g.*, on OlympiadBench and MathVision. But the gains are marginal relative to the overall model abilities and come at a

---

[1]Due to the length of trajectories and the use of sampling, duplicates are virtually impossible

| Training Data | | | MMMU | MathVista | MathVision | OlympiadBench | Average |
|---|---|---|---|---|---|---|---|
| GV | Text | RC | | | | | |
| ✓ | | | 59.67 | 69.20 | 31.28 | 24.95 | 46.28 |
| | ✓ | | 66.56 | 73.90 | 42.86 | 44.66 | 57.00 |
| | | ✓ | 65.44 | 70.90 | 42.73 | 41.46 | 55.13 |
| ✓ | ✓ | | 67.78 | 71.90 | 43.62 | 44.84 | 57.04 |
| ✓ | | ✓ | 65.66 | 70.70 | 43.27 | 43.19 | 55.71 |
| | ✓ | ✓ | 65.44 | 72.00 | 42.89 | 46.05 | 56.60 |
| ✓ | ✓ | ✓ | 66.33 | 74.70 | 47.96 | 49.22 | 59.55 |

Table A3: Pass@1 accuracy of SOPHIA in 38B scalre and baselines. "GV" and "Text" indicate the general visual and textual data, respectively, while "RC" represents the trajectory and update strategy from SOPHIA. A ✓ denotes that the corresponding data type is used in the experiment.

| Method | College-level Question | | Math-related Question | | | Challenging Scientific Reasoning | | | Average |
|---|---|---|---|---|---|---|---|---|---|
| | MMMU | MMMU Pro | MathVista | MathVerse | DynaMath | MathVision | MV-MATH | OlympiadBench | |
| Keep-1 | 65.44 | 46.93 | 70.90 | 45.18 | 59.14 | 42.73 | 34.15 | 41.46 | 50.74 |
| Keep-2 | 64.22 | 45.83 | 72.40 | 44.67 | 60.34 | 42.82 | 30.81 | 43.91 | 50.63 |
| Keep-8 | 64.56 | 45.61 | 72.00 | 45.95 | 61.22 | 44.87 | 31.71 | 45.71 | 51.45 |

Table A4: Pass@1 accuracy of SOPHIA on InternVL 2.5 38B in different $N$ ($N = 1, 2, 8$).

higher computational cost. Therefore, considering the trade-off between performance and efficiency, we adopt the Keep-1 strategy.

# E  Prompt

**Image Caption Prompt.** Figure A2 is the prompt for describing images to provide comprehensive captions including objects, color, text and so on. We do not reveal the question associated with the image to the vision model, as doing so may prompt it to answer the question directly, leading to potential reward hacking when evaluating captions.

**Simulating Visual Reasoning Prompt.** Figure A3 is the prompt for enabling the reasoning model to simulate a LVLM that can see the image based on provided caption. We build sample responses for the model, and they are typically generated automatically and then refined manually with additional visual operations (*e.g.*, "look back"), which enables in-context learning that produces replies with consistent formatting and greater suitability for visual tasks.

**Slow-thinking Chain-of-Thought Reasoning.** Figure A4 is the system prompt for the slow-thinking LVLM. During training, we use this system prompt when the trajectory contains a <think>...</think> tag; otherwise, the default system prompt is applied. This approach helps the model adapt to data from different distributions. At inference and evaluation time, we consistently apply this system prompt to our model.

**Answer Extracting Prompt.** Figure A5 is the prompt for extracting answer from the response. Examples are provided to prevent the LLM from answering questions instead of extracting the original answer.

**Answer Evaluation Prompt.** Figure A6 is the prompt for evaluating whether the answer is correct.

# F  Limitations

Despite the effectiveness of SOPHIA in enhancing visual slow-thinking reasoning, several limitations remain. First, LVLMs still struggle to retain long-range visual dependencies, especially in multi-hop or complex reasoning tasks, which can lead to partial or fragmented understanding of intricate visual scenes. Second, the current visual encoder lacks sufficient fine-grained recognition capability for complex scenes. We acknowledge that some low-level features, such as gradients and pixels, are not captured by semantic representations, since they are largely independent of semantics, which restricts the fidelity of visual perception and, consequently, the precision of downstream reasoning tasks that rely heavily on such features. Third, although the reward design in SOPHIA effectively filters out flawed trajectories, persistent issues such as hallucination and reasoning redundancy are not fully resolved, as evidenced by occasional overthinking or repetitive outputs in challenging

benchmarks. Addressing these challenges in future work could involve integrating stronger visual grounding techniques, adaptive trajectory pruning, and multi-stage curriculum learning to further improve long-range visual reasoning and visual fidelity.

# G    Broader Impacts

This paper presents SOPHIA, a semi-off-policy reinforcement learning framework which advances vision-language models by improving both visual understanding and reasoning without relying on human or closed-source annotations. By enabling scalable, automated training and providing open evaluation benchmarks, SOPHIA supports the development of more reliable and generalizable AI systems for complex multimodal tasks. We hope our open resources and methods will foster further research and broader real-world impact, particularly in education, science, and assistive technologies.

**Image Caption Prompt:**
Provide a highly detailed and precise description of every visible element within the image, capturing all visual and textual aspects exactly as they appear.

For images containing text, thoroughly extract and transcribe all visible words, symbols, numbers and time, including labels, annotations, legends, and numerical values. Pay close attention to font style, size, color, and positioning of each text element. Describe any formatting or emphasis, such as bold or italic text, and note any alignment or proximity between text elements.

For the connections between nodes, describe the type of line or arrow used, its direction, and any associated labels, symbols, or annotations that clarify the meaning of the relationship. If there are multiple types of connections, make sure to distinguish between them and describe how they are represented visually (e.g., solid vs. dashed lines, different colors, or arrowheads). Provide context for the meaning of each connection, especially in terms of the relationships or interactions they represent between the nodes.

If the image includes any type of statistical chart or graph, such as bar charts, line charts, scatter plots, pie charts, or any other type of diagram, describe each element in detail. For geometric shapes, lines, points, or bars, specify their labels, colors, positions, and exact measurements or values when visible. For bar charts, include each bar's label, color, height or value, category labels, legends, and axis labels, carefully noting the numerical scale on the axes and any increments or intervals. For line charts, detail the data points, lines, their color, and any markers or trends, indicating the specific values at key positions, note that every data point should be interpreted with numerical value. For scatter plots, describe each point's position, color, and any patterns or clusters. For pie charts, describe each segment's label, color, proportion, and the percentage or value it represents. Always ensure that any numerical scale, axis labels, and unit measurements are clearly noted, including any legends or other graphical annotations that may aid in understanding the data.

In complex scenes, describe every object in exhaustive detail, thoroughly extract and transcribe all visible words, symbols, numbers and time, noting its color, texture, shape, size, and relative position within the scene. Start with the objects in the foreground and gradually move towards the background, maintaining a clear and logical sequence. Provide information on spatial arrangements, including any overlapping, layering, or alignment among objects. Mention any shadows, highlights, or lighting effects that influence the appearance of objects. When describing the background, include its color, patterns, or gradients, and any additional visual elements that contribute to the scene, ensuring that the focus shifts progressively from the front to the back. This methodical, from-front-to-back approach will help provide a coherent and structured visualization of the scene.

If the image depicts realistic human figures, provide a detailed description of each individual's posture, physical appearance, clothing, facial expressions, and interactions with the surroundings. If the image includes well-known figures, make every effort to identify them by their facial features, attire, or contextual clues. If identification is uncertain, describe potential characteristics or identity hints in detail.

Lastly, aim to describe the image as comprehensively as possible, ensuring no detail is overlooked. Capture every nuance in the image, providing an accurate, in-depth account of each element and its precise attributes, arrangement, and interactions within the scene.

Figure A2: Prompts for comprehensively captioning images.

**Simulating Visual Reasoning Prompt:**
Please read the following example carefully.

<Begin of example>
Question: {The example of visual question}
Visual information:
{The example of image caption}

Your response:
{The example of response from reasoning model}
<End of example>

Offer a comprehensive breakdown of your analytical process based on the given visual information, detailing each step, the reasoning behind your decisions, and how you integrated various pieces of information. If you find you are overthinking this, you can stop analysis and determine the final answer. If you can directly find answer from visual information, just copy the original text, and no need to explain what is the meaning of the text or provide more context. Present your final answer at the end after "**Final answer**" and wrapped with boxed{}. You should assume that you can see this image, and you must use 'image' to refer to the visual information.

Question: {Provided visual question}
Visual information:
{Provided image caption}

Your response:

Figure A3: Prompts for simulating LVLM to answer questions.

**System Prompt:**
You are an expert mathematician with extensive experience in mathematical competitions. You approach problems through systematic thinking and rigorous reasoning. When solving problems, follow these thought processes:

## Deep Understanding
Take time to fully comprehend the problem before attempting a solution. Consider:
- What is the real question being asked?
- What are the given conditions and what do they tell us?
- Are there any special restrictions or assumptions?
- Which information is crucial and which is supplementary?
## Multi-angle Analysis
Before solving, conduct through analysis:
- What mathematical concepts and properties are involved?
- Can you recall similar classic problems or solution methods?
- Would diagrams or tables help visualize the problem?
- Are there special cases that need separate consideration?
## Systematic Thinking
Plan your solution path:
- Propose multiple possible approaches
- Analyze the feasibility and merits of each method
- Choose the most appropriate method and explain why
- Break complex problems into smaller, manageable steps
## Rigorous Proof
During the solution process:
- Provide solid justification for each step
- Include detailed proofs for key conclusions
- Pay attention to logical connections
- Be vigilant about potential oversights
## Repeated Verification
After completing your solution:
- Verify your results satisfy all conditions
- Check for overlooked special cases
- Consider if the solution can be optimized or simplified
- Review your reasoning process

Remember:
1. Take time to think thoroughly rather than rushing to an answer
2. Rigorously prove each key conclusion
3. Keep an open mind and try different approaches
4. Summarize valuable problem-solving methods
5. Maintain healthy skepticism and verify multiple times

Your response should reflect deep mathematical understanding and precise logical thinking, making your solution path and reasoning clear to others. When you're ready, present your complete solution with:
- Clear problem understanding
- Detailed solution process
- Key insights
- Thorough verification

Please put your thinking process within <think>...</think> tags.
Provide answers in the same language as the user asking the question. You have **32768** tokens to complete the answer.

Figure A4: The system prompt for long CoT reasoning.

**Answer Extracting Prompt:**
Please read the following example. Then extract the answer from the model response and type it at the end of the prompt.

Hint: Please answer the question requiring an integer answer and provide the final value, e.g., 1, 2, 3, at the end.
Question: Which number is missing?

Model response: The number missing in the sequence is 14.

Extracted answer: 14

Hint: Please answer the question requiring a floating-point number with one decimal place and provide the final value, e.g., 1.2, 1.3, 1.4, at the end.
Question: What is the fraction of females facing the camera?

Model response: The fraction of females facing the camera is 0.6, which means that six out of ten females in the group are facing the camera.

Extracted answer: 0.6

Hint: Please answer the question requiring a Python list as an answer and provide the final list, e.g., [1, 2, 3], [1.2, 1.3, 1.4], at the end.
Question: Between which two years does the line graph saw its maximum peak?

Model response: The line graph saw its maximum peak between 2007 and 2008.

Extracted answer: [2007, 2008]

Hint: {Provided hint}
Question: {Provided question}

Model response: {Provided response from the model}

Extracted answer:

Figure A5: Prompts for extracting the answer from response.

**Answer Evaluation Prompt:**
Below are two answers to a math question. Question is [Question], [Standard Answer] is the standard answer to the question, and [Model_answer] is the answer extracted from a model's output to this question. Determine whether these two answers are consistent.
Please note that only when the [Model_answer] completely matches the [Standard Answer] means they are consistent. For non-multiple-choice questions, if the meaning is expressed in the same way, it is also considered consistent, for example, 0.5m and 50cm.
If they are consistent, Judgement is 1; if they are different, Judgement is 0.

[Question]: Write the set of numbers represented on the number line in interval notation.
[Standard Answer]: (-2,1]
[Model_answer]: Extracted Answer: ((-2, 1))
Judgement: 0

[Question]: Find the domain and range of the function f using interval notation.
[Standard Answer]: domain: [-4, 0) and range: (-3, 1]
[Model_answer] : Range: ((-4, 1])
Judgement: 0

[Question]: Given the graph of the ellipse that intersects with x-axis at 9 and -9 and with y-axis at 3 and -3, determine its equation.A. $\frac{x^2}{81} + \frac{y^2}{9} = 1$ B. Can not determine.
[Standard Answer]: A
[Model_answer] : $\frac{x^2}{81} + \frac{y^2}{9} = 1$
Judgement: 1

[Question]: {question}
[Standard Answer]: {gt}
[Model_answer] : {extraction}
Judgement:

Figure A6: Prompts for evaluating whether the answer is correct.

