# OpenReview forum: "Semi-off-Policy Reinforcement Learning for Vision-Language Slow-Thinking Reasoning"
_NeurIPS.cc/2025/Conference — NeurIPS 2025 poster_

### Official Review · Reviewer_WfAw · 2025-06-22

**Clarity:** 3
**Significance:** 3
**Originality:** 3
**Rating:** 5
**Confidence:** 3

**Summary:**

This paper introduces SOPHIA, a novel Semi-Off-Policy Reinforcement Learning framework designed to enhance Large Vision-Language Models (LVLMs) with visual "slow-thinking" reasoning. It addresses the limitations of both on-policy RL's restricted exploration and off-policy RL's risk of visual hallucinations. SOPHIA proposes a semi-off-policy behavior model that integrates the LVLM's visual understanding with off-policy reasoning from external Language Models, using outcome-based and propagated visual rewards. Experiments across diverse visual reasoning benchmarks demonstrate SOPHIA's significant performance improvements for LVLMs like InternVL2.5 and InternVL3.0.

**Questions:**

Please refer to weakness.

**Ethical Concerns:**

["NO or VERY MINOR ethics concerns only"]

**Final Justification:**

Thank you for the detailed response. My concerns have been adequately addressed, and I would like to maintain my original positive score.

**Limitations:**

yes

**Quality:**

3

**Strengths And Weaknesses:**

Strengths
-  The paper clearly identifies a significant problem in LVLMs' slow-thinking reasoning and presents a well-motivated solution. The writing is notably clear, making the proposed methodology easy to understand.
-  The core design of SOPHIA is innovative and highly sensible, effectively tackling challenges faced by current RL approaches for LVLMs.
-  The work demonstrates high quality through comprehensive experimental validation and thorough ablation studies, providing robust support for its effectiveness.

Weaknesses
- The paper could benefit from a more detailed qualitative analysis of typical error modes before and after SOPHIA training, providing deeper insights into how visual hallucinations or logical inconsistencies are specifically mitigated.
- While SOPHIA is validated using InternVL2.5 and InternVL3.0, demonstrating its effectiveness when applied to other strong base LVLMs, such as Qwen-VL, as the core model for enhancement, would further strengthen the generalizability claims.
- The paper also lacks a detailed analysis of computational costs, covering both training phase resource consumption and the inference latency associated with multi-step "slow-thinking" reasoning.

---

> ### Author Rebuttal · Authors · 2025-07-31
>
> We thank Reviewer WfAw for your positive feedback. We are pleased to answer reviewer’s concerns below and will incorporate all feedback to polish up the paper.
>
> > **Review**: The paper could benefit from a more detailed qualitative analysis of typical error modes before and after SOPHIA training, providing deeper insights into how visual hallucinations or logical inconsistencies are specifically mitigated.
>
> **Response**: Thanks for the suggestion. We conduct additional manual evaluation on 50 randomly sampled cases from MMMU Pro before and after SOPHIA training, covering diverse disciplines to assess both visual hallucination and logical inconsistency. We report the proportion of responses with visual hallucinations and logic inconsistency. As shown in the table,  SOPHIA significantly reduces visual hallucinations and logic error, consistent with improvements in overall correctness.  This further supports our motivation that on-policy visual understanding training can enhance the model's visual understanding capabilities while reducing hallucinations.
>
> |Model|Visual Halluciation|Logic Inconsistancy|
> |---|---|---|
> |InternVL3.0-38B|38%|54%|
> |InternVL3.0-38B+SOPHIA|16%|44%|
>
> Due to space constraints, we are unable to provide specific qualitative examples in the rebuttal. We will present representative cases during the discussion phase and include these examples in the revised paper to better illustrate our motivation and the effectiveness of SOPHIA in reducing visual hallucinations.
>
> > **Review**: While SOPHIA is validated using InternVL2.5 and InternVL3.0, demonstrating its effectiveness when applied to other strong base LVLMs, such as Qwen-VL, as the core model for enhancement, would further strengthen the generalizability claims.
>
> **Response**: Thanks for this advice. Our method is not tied to a specific architecture and can be applied to other LVLM families with minimal modification. To quickly validate its generality, we further conduct research on Qwen2.5-VL in both 7B and 32B scales with a subset of the training data without warming-up and mixing data. The results on four benchmarks are shown in the following tables.
>
> ||MMMU|MathVista|MathVision|OlympiadBench|
> |---|---|---|---|---|
> |Qwen2.5-VL-7B|52.00|68.90|26.38|16.28|
> |Qwen2.5-VL-7B+SOPHIA|57.22|68.50|33.26|26.59|
> |Qwen2.5-VL-32B|65.33|72.80|38.98|38.92|
> |Qwen2.5-VL-32B+SOPHIA|68.22|73.70|43.68|41.99|
>
> We can observe that SOPHIA is also effective for the QwenVL family. We will present related results after revision.
>
> > **Review**: The paper also lacks a detailed analysis of computational costs, covering both training phase resource consumption and the inference latency associated with multi-step "slow-thinking" reasoning.
>
> **Response**: The sampling complexity of our method during the training phase is $O(MN)$, where $M$ represents the sampling size for visual understanding, and $N$ represents the sampling size for visual reasoning. During the entire training period, for the InternVL3.0-38B model we use, total GPU hours during the training phase are 2048 hours on A800 GPU. Note that for the same data scale, the GRPO training requires about 1536 hours on A800 GPU.
>
> For inference, the algorithm of the SOPHIA-trained model is no different from the base LVLM (next token prediction). However, its slow-thinking capability requires more tokens to complete the reasoning process. SOPHIA’s inference time is 1.64 GPU hours on A800 for MathVista benchmarks. For comparison, the inference cost for InternVL3.0-38B is 0.46 hours. Although SOPHIA introduces moderate additional inference latency compared to the base LVLM, this also reflects that SOPHIA is designed to cultivate slow-thinking, step-by-step reasoning, which is an ability absent from standard LVLMs. The increase in inference time is comparable to that observed in other recent reasoning LVLMs and LLMs, and follows the paradigm of test-time scaling in reasoning models. We believe this latency trade-off is reasonable given the significant gains in complex reasoning.
>
> We will revise our paper to make it more clear.

---

> > ### Comment · Reviewer_WfAw · 2025-08-01
> >
> > Thank you for your detailed and thoughtful response. I have carefully considered your clarifications, which effectively addressed my concerns. I appreciate your efforts, and my overall impression of the work remains positive.

---

> > > ### Author Response · Authors · 2025-08-01
> > > **Thanks to Reviewer WfAw**
> > >
> > > We greatly appreciate your constructive feedback and your support for our work, and your high regard for our efforts is truly encouraging. We will carefully revise the paper according to your suggestions and believe that these revisions have greatly improved the paper. Thank you once again.

---

### Official Review · Reviewer_F2hY · 2025-07-02

**Clarity:** 2
**Significance:** 3
**Originality:** 3
**Rating:** 4
**Confidence:** 3

**Summary:**

This paper introduces SOPHIA, a novel reinforcement learning framework designed to enhance visual slow-thinking reasoning capabilities in large vision-language models (LVLMs). The work proposes the semi-off-policy sampling and reward evaluation and propagation to address a critical limitation in current LVLM training approaches for complex multimodal reasoning tasks. The method also achieves competitive performance across several benchmarks.

**Questions:**

See weaknesses.

**Ethical Concerns:**

["NO or VERY MINOR ethics concerns only"]

**Final Justification:**

The author's response  answers most of my confusion. I will increase my rate to borderline accept.

**Limitations:**

Yes

**Paper Formatting Concerns:**

The paper has almost no formatting issues.

**Quality:**

3

**Strengths And Weaknesses:**

**Strengths**

-	The semi-off-policy proposed in the paper cleverly combines the advantages of on policy and off policy, which is a very interesting work.
-	The design of the reward mechanism is relatively novel, avoiding human evaluation and establishing effective end-to-end optimization, with good scalability
-	This method has achieved competitive results on different benchmarks

**Weaknesses**

-	The paper lacks a complete description of the process, especially regarding the training and inference processes. It is not easy to understand the complete workflow, for example, does the inference stage require a reasoning language model?
-	The design of the reward mechanism is based on the assumption that high-quality visual descriptions are strongly correlated with final results. Is there clear evidence to support this assumption?
-	The paper only validated InternVL as a VLM, and it is recommended to try other VLMs as this method has been proven to be universal for different VLM models.
-	Does SOPHIA have advantages over direct distillation data? For example, first generate capture data using VLM, and reasoning trajectories with reasoning language model, then build a dataset using reject sampling, and train the model using SFT.
-	It seems that the paper does not analyze the impact of different reasoning models on performance.
-	Suggest the author to provide a more detailed explanation of the fundamental difference between the proposed method and knowledge distillation.

---

> ### Author Rebuttal · Authors · 2025-07-31
>
> We thank Reviewer F2hY for your comprehensive feedback. We are happy to address the reviewer’s concerns and will incorporate all feedback to improve the paper.
>
> > **Review**: The paper lacks a complete description of the process, especially regarding the training and inference processes. It is not easy to understand the complete workflow, for example, does the inference stage require a reasoning language model?
>
> **Response**: Our paper provides a detailed description of the SOPHIA workflow in both training and inference stages. As shown in Figure 2 and Section 4, SOPHIA combines on-policy visual understanding from the LVLM with off-policy reasoning from a language model during training (line 129-149) , and the reward design, as well as the policy updating, is explained in section 4.2 (line 150-168) and section 4.3 (line 170-176) respectively. During inference and evaluation, only the LVLM is used and no external reasoning language model is required (line 658-667), which is consistent with evaluation methods in other existing works. We will revise our paper to make it more clear.
>
> > **Review**: he design of the reward mechanism is based on the assumption that high-quality visual descriptions are strongly correlated with final results. Is there clear evidence to support this assumption?
>
> **Response**: Thanks for your question. Our reward mechanism is grounded in both recent studies and our own ablation analyses.
> First, recent work [1] shows that the adapter layers in LLaVA architecture LVLM act as mappings that project visual embeddings into subspaces spanned by corresponding text embeddings. This indicates that textualized visual information (i.e., high-quality visual descriptions) plays a central role in the following multimodal reasoning.
>
> Second, we conduct additional manual evaluation on 50 randomly sampled cases from MMMU Pro before and after SOPHIA training, covering diverse disciplines to assess both visual hallucination (including visual grounding error). We report the proportion of responses with visual hallucinations. As shown in the table,  SOPHIA significantly reduces visual hallucinations, consistent with improvements in overall correctness.  This further supports that reward mechanism based on the assumption is a key role in enhancing the model's visual understanding capabilities while reducing hallucinations.
>
> |Model|Visual Hallucinations|
> |---|---|
> |InternVL3.0-38B|38%|
> |InternVL3.0-38B+SOPHIA|16%|
>
> Finally, we empirically validate this assumption through ablation studies presented in  Section 6.3 (Table 5). Specifically, removing the visual description reward from SOPHIA leads to significant performance drops, particularly on challenging reasoning benchmarks such as MathVision and OlympiadBench. For example, on MathVision, removing the caption reward lowers the score from 42.73% to 40.35%. This shows that high-quality visual descriptions are indeed critical for guiding LVLMs toward accurate and robust reasoning.
>
> We will further clarify this motivation in the revised manuscript, and thank the reviewer for encouraging us to strengthen this discussion.
>
> > **Review**: The paper only validated InternVL as a VLM, and it is recommended to try other VLMs as this method has been proven to be universal for different VLM models.
>
> **Response**: Thanks for this advice. Our method is not tied to a specific architecture and can be applied to other LVLM families with minimal modification. To quickly validate its generality, we further conduct research on Qwen2.5-VL in both 7B and 32B scales with a subset of the training data without warming-up and mixing data. The results on four benchmarks are shown in the following tables.
>
> ||MMMU|MathVista|MathVision|OlympiadBench|
> |---|---|---|---|---|
> |Qwen2.5-VL-7B|52.00|68.90|26.38|16.28|
> |Qwen2.5-VL-7B+SOPHIA|57.22|68.50|33.26|26.59|
> |Qwen2.5-VL-32B|65.33|72.80|38.98|38.92|
> |Qwen2.5-VL-32B+SOPHIA|68.22|73.70|43.68|41.99|
>
> We can observe that SOPHIA is also effective for the QwenVL family. We will present related results after revision.
>
> > **Review**: Does SOPHIA have advantages over direct distillation data? For example, first generate capture data using VLM, and reasoning trajectories with reasoning language model, then build a dataset using reject sampling, and train the model using SFT.
>
> **Response**: We conduct extensive experiments in Table 1 and the ablation studies (Table 5), which show that SOPHIA consistently outperforms distillation data and reject sampling. In Table 1, SOPHIA achieves an average pass@1 accuracy of 55.51% on InternVL3.0-38B across eight benchmarks, compared to 49.51% with the MPO (extension of SFT). In Table 5, removing the caption reward from SOPHIA (SOPHIA w/o CR, i.e., reject sampling) leads to a notable decrease in average pass@1 accuracy on InternVL2.5-38B: from 55.13% (SOPHIA) to 53.54%. This also shows that SOPHIA's advantage over direct distillation data lies in its on-policy enhancement of visual understanding. This is particularly critical for LVLMs, as their visual understanding capabilities are often inherently limited, and direct distillation struggles to effectively improve this aspect. SOPHIA addresses this issue by enhancing the visual understanding of LVLMs on-policy, thereby more effectively improving multi-modal slow-thinking reasoning quality.
>
> > **Review**: It seems that the paper does not analyze the impact of different reasoning models on performance.
>
> **Response**: We present an analysis of the impact of different reasoning models in Appendix C.3 (Table A2). As shown, we vary the reasoning LLM and discuss their influence on SOPHIA’s performance. The results show that SOPHIA maintains robust gains across different reasoning model choices, indicating that SOPHIA can achieve strong performance without an extremely powerful teacher model.
>
> > **Review**: Suggest the author to provide a more detailed explanation of the fundamental difference between the proposed method and knowledge distillation.
>
> **Response**: We clarify the difference between SOPHIA and knowledge distillation in both the Introduction and Related Work sections, and address the strength of SOPHIA compared to distillation as aforementioned. While knowledge distillation passively transfers teacher outputs to a student, SOPHIA integrates on-policy visual understanding with off-policy reasoning and optimizes via reinforcement learning with propagated rewards. Our experiment results further highlight that SOPHIA improves multi-modal reasoning ability rather than mere pattern imitation, distinguishing it fundamentally from distillation-based approaches.
>
> ---
> **References**
>
> [1] Jiaqi Liao, et al. LangBridge: Interpreting Image as a Combination of Language Embeddings. In ICCV 2025.

---

> > ### Comment · Reviewer_F2hY · 2025-08-05
> >
> > Thank you for the author's response, which answered most of my confusion. I will increase my rate to borderline accept.

---

> > > ### Author Response · Authors · 2025-08-05
> > > **Thanks to Reviewer F2hY**
> > >
> > > We deeply appreciate your feedback. Should you have any additional questions or concerns, please feel free to raise them or discuss them at any time, and we will address them at the earliest opportunity. Your understanding and support are greatly valued.

---

### Official Review · Reviewer_oeeq · 2025-07-03

**Clarity:** 3
**Significance:** 3
**Originality:** 3
**Rating:** 4
**Confidence:** 2

**Summary:**

This paper proposes SOPHIA, a semi-off-policy reinforcement learning framework designed to enhance slow-thinking reasoning abilities in large vision-language models (LVLMs). SOPHIA combines the on-policy visual understanding of a trainable LVLM with off-policy reasoning trajectories generated by a language model, enabling more robust and scalable learning. Through extensive experiments on InternVL2.5 and InternVL3.0 (at both 8B and 38B scales), SOPHIA achieves state-of-the-art performance on multimodal reasoning tasks, outperforming both open-source and some closed-source models.

**Questions:**

In general, I believe this paper is solid and the results are impressive. I have some questions that needs further clarification

1. Can the authors provide qualitative examples or visualizations of improved visual grounding after SOPHIA training?
2. What is the inference latency introduced by using SOPHIA compared to the base LVLM?
3. Could SOPHIA be applied to other modalities or tasks beyond math reasoning?

**Ethical Concerns:**

["NO or VERY MINOR ethics concerns only"]

**Final Justification:**

I appreciate the additional experiments and encourage the authors to incorporate them in the revision. My concerns are properly addressed.

**Limitations:**

Yes

**Quality:**

3

**Strengths And Weaknesses:**

First of all, my research primarily focuses on 3D vision. RL and VLM are out of my expertise. Therefore, I can only provide a general assessment rather than insightful and critical review.

**Strengths:**

- SOPHIA’s semi-off-policy design smartly combines stable on-policy visual understanding with the flexibility of off-policy reasoning from a stronger language model. This setup addresses key issues like limited exploration in on-policy RL and hallucinations in off-policy RL.
- The method shows clear performance gains on tough multimodal benchmarks like MathVision and OlympiadBench, outperforming other open-source models and even GPT-4.1 in some cases.

**Weaknesses:**
- Analysis on visual hallucinations: While SOPHIA aims to reduce hallucinations through visual reward propagation, qualitative examples or failure modes would be benficial to strengthen the claim.
- Binary reward signal: As acknowledged in the limitation section, the reward mechanism is based on a simple binary outcome and a preference for the shortest correct trajectory. This is a coarse signal that may not capture the nuances of a high-quality reasoning process and could prematurely penalize more complex but valid reasoning paths.
- Limited generalization in some cases: The results show that performance can degrade on certain tasks or data distributions not well-represented in the training data (e.g., MathVista for the 8B model). This suggests that while powerful, the method's robustness may be sensitive to the domain and quality of the reasoning trajectories it learns from.

---

> ### Author Rebuttal · Authors · 2025-07-31
>
> We thank the Reviewer oeeq for your positive feedback. We are happy to address the reviewer’s concerns and will incorporate all feedback to improve the paper.
>
> > **Review**: Analysis on visual hallucinations: While SOPHIA aims to reduce hallucinations through visual reward propagation, qualitative examples or failure modes would be benficial to strengthen the claim. Can the authors provide qualitative examples or visualizations of improved visual grounding after SOPHIA training? (**Cons1&Q1**)
>
> **Response**: Thanks for the suggestion. We conduct additional manual evaluation on 50 randomly sampled cases from MMMU Pro before and after SOPHIA training, covering diverse disciplines to assess both visual hallucination (including visual grounding error). We report the proportion of responses with visual hallucinations. As shown in the table,  SOPHIA significantly reduces visual hallucinations, consistent with improvements in overall correctness.  This further supports our motivation that on-policy visual understanding training can enhance the model's visual understanding capabilities while reducing hallucinations.
>
> |Model|Visual Hallucinations|
> |---|---|
> |InternVL3.0-38B|38%|
> |InternVL3.0-38B+SOPHIA|16%|
>
> Due to space constraints, we are unable to provide specific qualitative examples in the rebuttal. We will present representative cases during the discussion phase and include these examples in the revised paper to better illustrate our motivation and the effectiveness of SOPHIA in reducing visual hallucinations.
>
> > **Review**: Binary reward signal: As acknowledged in the limitation section, the reward mechanism is based on a simple binary outcome and a preference for the shortest correct trajectory. This is a coarse signal that may not capture the nuances of a high-quality reasoning process and could prematurely penalize more complex but valid reasoning paths.
>
> **Response**: We would like to first denote that our training supervision is not inherently limited to binary outcome rewards. With the fine-grained reward signal for visual understanding computed by reward propagation strategies, it is possible to employ more detailed feedback to the visual understanding during the visual reasoning process.
>
> Additionally, we agree that for the reasoning stage, the binary reward signal is a simplification. This design choice follows standard practice in current reasoning model training (e.g., DeepSeek-R1) [1], which is motivated by the challenges of reliably assigning stepwise or token-level rewards in complex reasoning. Specifically, some works propose using such process reward design [2], but fails in terms of long Chain-of-Thought due to value initialization bias and reward signal decay [3]. While outcome-based rewards may lack granularity, they prove effective in incentivizing correct reasoning without introducing substantial noise. We agree that exploring richer reward signals is important future work but is beyond this paper's scope.
>
> > **Review**: Limited generalization in some cases: The results show that performance can degrade on certain tasks or data distributions not well-represented in the training data (e.g., MathVista for the 8B model). This suggests that while powerful, the method's robustness may be sensitive to the domain and quality of the reasoning trajectories it learns from.
>
> **Response**: We appreciate this observation. We agree that model performance may decline on tasks underrepresented in the training data, but this issue is significantly alleviated as the model size increases. In Table 1, the performance of the 38B model on MathVista still surpasses these base models. Therefore, we acknowledge that data distribution can influence the effectiveness of the method to some extent, especially for smaller models, which are more sensitive to data during fine-tuning, but for larger models, this issue is greatly mitigated.
>
> > **Review**: What is the inference latency introduced by using SOPHIA compared to the base LVLM?
>
> **Response**: The inference algorithm of SOPHIA is no different from the base LVLM. However, its slow-thinking capability requires more tokens to complete the reasoning process. SOPHIA’s inference time is 1.64 GPU hours on A800 for MathVista benchmarks. For comparison, the inference cost for InternVL3.0-38B is 0.46 hours. Although SOPHIA introduces moderate additional inference latency compared to the base LVLM, this also reflects that SOPHIA is designed to cultivate slow-thinking, step-by-step reasoning, which is an ability absent from standard LVLMs. The increase in inference time is comparable to that observed in other recent reasoning LVLMs and LLMs, and follows the paradigm of test-time scaling in reasoning models. We believe this latency trade-off is reasonable given the significant gains in complex reasoning.
>
> > **Review**: Could SOPHIA be applied to other modalities or tasks beyond math reasoning?
>
> **Response**: Thanks for this question. While our main focus is on mathematical and scientific reasoning, the SOPHIA framework is modality- and domain-agnostic: it can in principle be extended to other tasks that require complex, step-wise reasoning, such as audio reasoning. In our experiments, we also evaluate SOPHIA on benchmarks such as MMMU and MMMU Pro, which cover wide range of fields beyond mathematics, including law, medicine, art, social science and engineering, and observe consistent performance gains (46.47->52.60 on MMMU Pro for InternVL3.0-38B).  We believe SOPHIA provides general improvement in slow-thinking reasoning.
>
> ---
> **References**
>
> [1] Zhihong Shao, et al. DeepSeekMath: Pushing the Limits of Mathematical Reasoning in Open Language Models. arXiv 2402.03300.
>
> [2] Hunter Lightman, et al. Let's verify step by step. In ICLR 2023.
>
> [3] Yufeng Yuan, et al. What’s Behind PPO’s Collapse in Long-CoT? Value Optimization Holds the Secret. arXiv 2503.01491.

---

> > ### Comment · Reviewer_oeeq · 2025-08-04
> > **Thanks for the rebuttal!**
> >
> > I have read other reviews and rebuttals carefully. I appreciate the additional experiments and encourage the authors to incoporate them in the revision.  My concerns are properly addressed. I would like to keep my original score.

---

> > > ### Author Response · Authors · 2025-08-05
> > > **Thanks to Reviewer oeeq**
> > >
> > > Thank you for your valuable feedback. If you have any further questions or concerns, please do not hesitate to bring them up at your convenience, and we will promptly address them. We greatly appreciate your understanding and support, and will incorporate your suggestions into the paper to further polish it up.

---

### Official Review · Reviewer_xBLR · 2025-07-06

**Clarity:** 3
**Significance:** 2
**Originality:** 2
**Rating:** 4
**Confidence:** 4

**Summary:**

This paper proposes a semi-off-policy reinforcement learning framework to
enhance slow-thinking reasoning in large vision-language models. The approach
works by combining on-policy visual captioning from a trainable LVLM with
off-policy slow-thinking trajectories generated by a reasoning LLM. Visual and
reasoning rewards are computed and used to fine-tune the LVLM using off-policy
RL. Experiments on different benchmarks demonstrate the effectiveness of the
proposed method.

**Questions:**

See Weaknesses.

**Ethical Concerns:**

["NO or VERY MINOR ethics concerns only"]

**Final Justification:**

Most of my concerns are addressed.

**Limitations:**

yes

**Quality:**

3

**Strengths And Weaknesses:**

# Strengths

* The studied problem is important and interesting.

* The motivation for improving the on-policy and off-policy LLM RL is clear.

* The proposed method outperforms supervised fine-tuning and direct on-policy RL methods

# Weaknesses

* A central assumption in this work is that image content relevant to reasoning
tasks can be effectively captured through textual descriptions. While this may
hold for certain structured scenes, it overlooks a fundamental difference
between modalities: images are continuous and often encode fine-grained spatial
and visual information that discrete language space may fail to represent fully.
Relying entirely on language-based captions generated by an MLLM risks omitting
critical visual cues, especially in tasks where spatial relations or low-level
visual details influence the reasoning outcome. This paper would benefit from a
more explicit discussion of this and an analysis of how potential information
loss in the captioning step might affect downstream performance.

* The current formulation of the proposed method appears particularly
well-suited for tasks in which visual inputs already contain structured symbolic
content, such as text-labeled diagrams or schematic illustrations. In these
cases, the conversion from image to language retains most of the task-relevant
information, enabling the reasoning LLM to perform effectively. However, for
scenarios that require more perceptual visual understanding, e.g., interpreting
natural images, or abstract style and spatial patterns, this kind of assumption
may no longer hold. This limitation is also reflected in the choice of
evaluation benchmarks: most of the datasets used (e.g., MathVision,
OlympiadBench, DynaMath) focus on math, science, or diagram-based reasoning
tasks where textual abstraction is relatively reliable. There is a lack of
evaluation on more diverse, perception-centric benchmarks that stress visual
grounding beyond captionable content.

* The performance gains achieved by SOPHIA are largely the result of
incorporating reasoning trajectories from stronger language-only models such as
QwQ-32B-Preview and DeepSeek-R1. Since QwQ and R1 has much stronger inherent
reasoning capabilities than InternVL, part of the improvement may come simply
from distilling a more capable teacher. While this is a valid and useful
strategy, it complicates the interpretation of the results: it is difficult to
disentangle the contribution of the semi-off-policy framework itself from that
of the teacher model’s strength. An ablation study using different teacher
models of varying quality or complexity could help clarify this point.

* Another limitation relates to the training data used to support the method’s
performance. A significant portion of the results is based on a private dataset,
which is not released or described in sufficient detail. While the authors take
care to evaluate SOPHIA on public benchmarks as well, the strongest gains appear
on the internal dataset, whose construction, domain coverage, and difficulty
level remain unspecified. This makes it challenging to fully assess the
generality of the results or to reproduce them independently. More transparency
on dataset characteristics or representative examples would strengthen the
contribution.

* Additionally, the entire experimental study is conducted on InternVL models
only, across both 8B and 38B scales. While InternVL is a strong baseline, it
would be valuable to test the framework on other MLLM families with different
architectures or pretraining approaches.

* What is the time and computation resouces overhead from the proposed method?

---

> ### Author Rebuttal · Authors · 2025-07-31
>
> We thank the Reviewer xBLR for your comprehensive feedback. We are delighted to answer reviewer's concerns below to refine our paper.
>
> > **Review**: The paper assumes that textual descriptions can fully capture image content for reasoning tasks, but this may overlook fine-grained visual details inherent in images, potentially leading to information loss that warrants further discussion and analysis.
>
> **Response**: Our work builds upon the LLaVA architecture LVLM, which aligns visual features with language semantics.  Some studies [1] show that adapter layers project visual embeddings into subspaces spanned by corresponding text embeddings, supporting our core assumption. While converting images to language inevitably loses some low-level details, some elements such as color, shape, texture, and layout have natural language annotations and can be implicitly encoded within the semantic representations during pretraining [2, 3].
>
> Therefore, as shown in Figure A2 (page 22), we can use deliberately designed prompts to guide the LVLM to obtain the most comprehensive visual descriptions covering object color, texture, shape, size, spatial relationships, foreground/background, and object interactions within the scene. The "critical visual cues including spatial relations or low-level visual details" you mentioned are, for the most part, addressed by our instructions.
>
> Our experiments also show that the model achieves noticeable improvements in spatial reasoning and visual understanding (see our second response for more details). We will further revise the paper to discuss this issue and provide a more detailed analysis of how potential information loss in the captioning step may affect downstream performance.
>
> > **Review**: The proposed method excels at tasks involving structured, text-labeled visual inputs where image-to-language conversion preserves essential information, but its effectiveness is limited for perception-driven scenarios, as reflected by the evaluation's focus on diagram-based rather than perceptual benchmarks.
>
> **Response**: We would like to emphasize, as we mentioned in response to the previous question, that our framework is also capable of capturing some low-level visual features by the deliberately designed prompt. Some benchmarks we evaluated in this work also require perception abilities of low-level visual features. For example, in MMMU Pro, many subsets such as art, history and design comprehensively assess visual grounding and perception, on which SOPHIA improves InternVL 3.0 38B pass@1 acc from 46.47 to 52.60. Additionally, the MathVision you mentioned contains numerous spatial reasoning tasks (e.g., identifying the side view of a given cube in combinatorial geometry). And on this kind of tasks, the untrained InternVL 3.0 38B achieves an average of 21.75 pass@1 acc,  while the SOPHIA-trained model reaches 32.79, showing a significant improvement. We will explore more perception-centric visual slow-thinking reasoning in future studies.
>
> > **Review**: The observed performance gains of SOPHIA may be primarily due to distillation from stronger language-only teacher models rather than the semi-off-policy framework itself, making it challenging to isolate the framework’s true contribution without further ablation studies using teachers of varying quality.
>
> **Response**: We agree that SOPHIA leverages the reasoning capabilities of strong language-only teacher models. However, as shown in our ablation studies (Table 5), semi-off-policy framework contributes independently to performance gains (line SOHPIA), above and beyond raw teacher strength (line SOHPIA w/o CR). We further explore different teacher models in Appendix C.3 (Table A2), finding that no significant differences between two different models in terms of quality (DeepSeek > QwQ-32B-Preview) and size (658B > 32B), indicating that SOPHIA can achieve strong performance without an extremely powerful teacher model.
>
> > **Review**: The reliance on an unreleased and insufficiently described private dataset, which accounts for the strongest results, limits the reproducibility and generalizability of the findings, highlighting the need for greater transparency regarding dataset characteristics.
>
> **Response**: Indeed, we use internal datasets to train the model, as currently available large-scale visual reasoning datasets are generally of low quality. Our dataset is composed of diverse and challenging multimodal questions from K12 and higher education, spanning science, math, finance, and social sciences, and is carefully curated to ensure a balanced difficulty distribution and prevent data leakage.
>
> We also conduct experiments on public visual datasets, including MathV360K (mentioned in paper), M3CoT, llava-CoT, etc., but due to issues such as lower difficulty levels and limited diversity in question types in these datasets, the performance was not as strong as with our internal dataset. Despite these weaknesses of public datasets, SOPHIA still shows improvements (Table 6) and the public datasets are competitive with private datasets. To support reproducibility, we will release our internal training dataset upon acceptance.
>
> > **Review**:  Additionally, the entire experimental study is conducted on InternVL models only, across both 8B and 38B scales. While InternVL is a strong baseline, it would be valuable to test the framework on other MLLM families with different architectures or pretraining approaches.
>
> **Response**: Thanks for this advice. Our method is not tied to a specific architecture and can be applied to other LVLM families with minimal modification. To quickly validate its generality, we further conduct research on Qwen2.5-VL in both 7B and 32B scales with a subset of the training data without warming-up and mixing data. The results on four benchmarks are shown in the following tables.
>
> ||MMMU|MathVista|MathVision|OlympiadBench|
> |---|---|---|---|---|
> |Qwen2.5-VL-7B|52.00|68.90|26.38|16.28|
> |Qwen2.5-VL-7B+SOPHIA|57.22|68.50|33.26|26.59|
> |Qwen2.5-VL-32B|65.33|72.80|38.98|38.92|
> |Qwen2.5-VL-32B+SOPHIA|68.22|73.70|43.68|41.99|
>
> We can observe that SOPHIA is also effective for the QwenVL family. We will present related results after revision.
>
> > **Review**: What is the time and computation resouces overhead from the proposed method?
>
> **Response**: The sampling complexity of our method during the training phase is $O(MN)$, where $M$ represents the sampling size for visual understanding, and $N$ represents the sampling size for visual reasoning. During the entire training period, for the InternVL3.0-38B model we use, total GPU hours during the training phase are about 2048 hours on A800 GPU. Note that for the same data scale, the GRPO training requires about 1536 hours on A800 GPU.
>
> ---
> **References**
>
> [1] Jiaqi Liao, et al. LangBridge: Interpreting Image as a Combination of Language Embeddings. In ICCV 2025.
>
> [2] Rui Xiao, et al. FLAIR: VLM with Fine-grained Language-informed Image Representations. In CVPR 2025.
>
> [3] Michael Tschannen, et al. SigLIP 2: Multilingual Vision-Language Encoders with Improved Semantic Understanding, Localization, and Dense Features. arXiv 2502.14786.

---

> > ### Comment · Reviewer_xBLR · 2025-08-09
> >
> > Thanks for the detailed response. Most of my concerns are addressed. I will increase my score accordingly.

---

> ### Author Response · Authors · 2025-08-07
> **Kind Request for Your Feedback on Our Rebuttal**
>
> Thank you for your recognition of our paper, as well as for the valuable questions and suggestions you have provided. During both the rebuttal and discussion phases, we do our best to address all the concerns and issues you raised. If you have any additional questions or concerns, please feel free to bring them up at any time, and we will respond as soon as possible. If our response has resolved your concerns and you are satisfied with our reply, we would sincerely appreciate your consideration in raising our score. Thank you very much.

---

### Author Response · Authors · 2025-08-01
**Paper Clarification (1/2)**

We would like to express our thanks to reviewers for their thorough reading of the paper and insightful comments. We at first clarify some most common questions and concerns raised by reviewers to avoid any misunderstandings.

**1. Assumption of Textual Representation for Visual Content and Reasoning**

The SOPHIA framework in our paper is built on the assumption that textual descriptions can capture the critical visual cues relevant to reasoning tasks. This assumption is supported by recent studies as well as our own experimental analyses.

First, our work is based on LVLMs using the LLaVA architecture. Recent research [1] shows that the adapter layers in LLaVA architecture LVLM act as mappings that project visual embeddings into subspaces spanned by corresponding text embeddings. This suggests that the continuous visual features are similar to the textualized visual information within the semantic space of LLaVA models. Thus, **we can consider the visual understanding** (whether continuous features or discrete textual tokens) **as the condition for subsequent autoregressive text generation**. If this "condition" includes accurate and comprehensive visual cues, the downstream reasoning is more likely to be logically concistant.

Additionally, we acknowledge that some low-level features, such as gradients and pixels, are not captured by semantic representations, since they are largely independent of semantics, which also indicates the inherent limitations in tasks that rely heavily on such features. However, **many critical low-level visual cues, such as object color, texture, shape, size, and spatial relationships, are richly annotated with semantics, and can be implicitly encoded within the semantic representations during alignment and pretraining** [2, 3]. Therefore, we can use deliberately designed prompts to guide the LVLM to obtain the most comprehensive visual descriptions covering them, as shown in Figure A2 (page 22).

Because these features can be captured to a certain extent, our experiments show that the SOPHIA-trained model achieves significant improvements not only on tasks where the visual input already contains structured symbolic content, but also on tasks involving spatial reasoning and visual understanding. For example, in MMMU Pro, many subsets such as art, history and design comprehensively assess visual grounding and perception, on which SOPHIA improves InternVL 3.0 38B pass@1 acc from 46.47 to 52.60. Additionally, the MathVision contains numerous spatial reasoning tasks (e.g., identifying the side view of a given cube in combinatorial geometry). And on this task, the untrained InternVL 3.0 38B achieves an average of 21.75 pass@1 acc,  while the SOPHIA-trained model reaches 32.79, showing a significant improvement. These results further validate our initial assumption.

At the same time, our experiments also reveal cases where the model struggles due to information loss during the captioning step. We will include a detailed discussion of these cases in the final version of the paper. We sincerely appreciate the reviewers' suggestions to strengthen this discussion.

**2. Effectiveness of SOPHIA Framework**

We conduct extensive experiments to show the effectiveness of the SOPHIA framework (particularly the  semi-off-policy strategy and reward propagation). We agree that SOPHIA leverages the reasoning capabilities of strong language-only teacher models. However, as shown in our ablation studies (Table 5), semi-off-policy framework contributes independently to performance gains (line SOHPIA), above and beyond raw teacher strength (line SOHPIA w/o CR). We further explore different teacher models in Appendix C.3 (Table A2), finding that no significant differences between two different models in terms of quality (DeepSeek-R1 > QwQ-32B-Preview) and size (658B > 32B), indicating that SOPHIA can achieve strong performance without an extremely powerful teacher model.

---

> ### Author Response · Authors · 2025-08-01
> **Paper Clarification (2/2)**
>
> **3. Generality and Applicability to Other Base LVLMs**
>
> Our method is not tied to a specific architecture and can be applied to other LVLM families with minimal modification. To quickly validate its generality, we further conduct research on Qwen2.5-VL in both 7B and 32B scales with a subset of the training data without warming-up and mixing data. The results on four benchmarks are shown in the following tables.
>
> ||MMMU|MathVista|MathVision|OlympiadBench|
> |---|---|---|---|---|
> |Qwen2.5-VL-7B|52.00|68.90|26.38|16.28|
> |Qwen2.5-VL-7B+SOPHIA|57.22|68.50|33.26|26.59|
> |Qwen2.5-VL-32B|65.33|72.80|38.98|38.92|
> |Qwen2.5-VL-32B+SOPHIA|68.22|73.70|43.68|41.99|
>
> We can observe that SOPHIA is also effective for the QwenVL family. We will present related results after revision.
>
> **4. Qualitative Analysis and Insights on the Mitigation of Visual Hallucinations and Logical Inconsistencies**
>
> We conduct additional manual evaluation on 50 randomly sampled cases from MMMU Pro before and after SOPHIA training, covering diverse disciplines to assess both visual hallucination and logical inconsistency. We report the proportion of responses with visual hallucinations and logic inconsistency. As shown in the table,  SOPHIA significantly reduces visual hallucinations and logic error, consistent with improvements in overall correctness.  This further supports our motivation that on-policy visual understanding training can enhance the model's visual understanding capabilities while reducing hallucinations.
>
> ||Visual Halluciation|Logic Inconsistency|
> |---|---|---|
> |InternVL3.0-38B|38%|54%|
> |InternVL3.0-38B+SOPHIA|16%|44%|
>
> In the discussion section, we present an example to illustrate the model’s responses before and after training. As shown, the model not only makes more accurate observations of the image but also avoids certain logical errors by repeatedly verifying its reasoning process. We will also include more examples in the revised paper to better illustrate our motivation and the effectiveness of SOPHIA in reducing visual hallucinations.
>
> **5. Computational Costs of Training and Inference**
>
> The sampling complexity of our method during the training phase is $O(MN)$, where $M$ represents the sampling size for visual understanding, and $N$ represents the sampling size for visual reasoning. During the entire training period, for the InternVL3.0-38B model we use, total GPU hours during the training phase are 2048 hours on A800 GPU. Note that for the same data scale, the GRPO training requires about 1536 hours on A800 GPU.
>
> For inference, the algorithm of the SOPHIA-trained model is no different from the base LVLM (next token prediction). However, its slow-thinking capability requires more tokens to complete the reasoning process. SOPHIA’s inference time is 1.64 GPU hours on A800 for MathVista benchmarks. For comparison, the inference cost for InternVL3.0-38B is 0.46 hours. Although SOPHIA introduces moderate additional inference latency compared to the base LVLM, this also reflects that SOPHIA is designed to cultivate slow-thinking, step-by-step reasoning, which is an ability absent from standard LVLMs. The increase in inference time is comparable to that observed in other recent reasoning LVLMs and LLMs, and follows the paradigm of test-time scaling in reasoning models. We believe this latency trade-off is reasonable given the significant gains in complex reasoning. We will revise our paper to make it more clear.
>
> ---
> **References**
>
> [1] Jiaqi Liao, et al. LangBridge: Interpreting Image as a Combination of Language Embeddings. In ICCV 2025.
>
> [2] Rui Xiao, et al. FLAIR: VLM with Fine-grained Language-informed Image Representations. In CVPR 2025.
>
> [3] Michael Tschannen, et al. SigLIP 2: Multilingual Vision-Language Encoders with Improved Semantic Understanding, Localization, and Dense Features. arXiv 2502.14786.

---

### Author Response · Authors · 2025-08-01
**Examples of Mitigation of Visual Hallucinations and Logical Inconsistencies**

Below is a real test sample from our MathVision benchmark, which involves spatial reasoning. This example shows that our method can both mitigate hallucinations and improve the model's spatial reasoning capabilities beyond the reasoning of structured symbolic content.

**Source**: MathVision Test id 48

**Question** Michael has two building blocks. Each building block is made up of two cubes glued together. Which figure can he not make using the blocks?

**Golden Answer**: B

**Input images can be referred in the original benchmark**.

---

> ### Author Response · Authors · 2025-08-01
> **Response from InternVL 3.0 38B**
>
> Michael has two building blocks, each made up of two cubes glued together. Let's analyze the options to determine which figure he cannot make:
>
> (A) This figure requires 4 cubes, which Michael can make with his two blocks.
>
> (B) This figure requires 4 cubes, which Michael can make with his two blocks.
>
> > There is a logical error here: the model only considers the number of cubes, overlooking the fact that the building blocks cannot be disassembled.
>
> (C) This figure requires 4 cubes, which Michael can make with his two blocks.
>
> (D) This figure requires 4 cubes, which Michael can make with his two blocks.
>
> (E) This figure requires 5 cubes, which Michael cannot make with his two blocks.
>
> > Here is hallucination. Although there are only four cubes in the image, the model incorrectly identifies five.
>
> Therefore, the correct answer is (E).
>
> > This leads to an incorrect final answer.

---

> ### Author Response · Authors · 2025-08-01
> **Response from InternVL 3.0 38B+SOPHIA**
>
> Okay, let's tackle this problem. So, Michael has two building blocks, and each block is made up of two cubes glued together. The question is asking which figure he cannot make using these blocks. The options are labeled A through E, and each option is a different arrangement of cubes.
>
> Looking at image1, it's a simple illustration of two building blocks, each made of two cubes. The way they are depicted is with each block consisting of two cubes stacked on top of each other. So, each block is a 2x1x1 structure. That makes sense. So, each block is like a vertical domino, two cubes high.
>
> Now, Michael has two such blocks. He needs to see which figure he cannot make by combining these two blocks. The figures A through E are various arrangements of cubes. Each figure is a different configuration, and we need to determine which one cannot be formed by placing these two vertical domino blocks together. Let me look at each option one by one. Let's start with option A.
>
> Option A: Looking at the image, it's an L-shape with four cubes. That's possible. For example, three cubes forming the corner and one attached to one of the arms. But wait, each block is two cubes. So, maybe he can make an L-shape by combining the two blocks in a certain way. For example, pair the two cubes of the foot, and pair the two cubes of the upright, forming an L. So, yes, that should be possible.
>
> > During the process, it reflects on the problem constraints to avoid logical errors, and corrects the previous mistakes.
>
> Option B: This figure is a pyramid-like structure with a base of 1x3 and one cube on top. The pyramid in option B has four cubes. If we have two blocks, each two cubes, that's a total of four cubes. Michael has four cubes total. Wait, hold on. Let me double-check. The pyramid in option B has a base of three cubes and one on top, making four cubes. Wait, if you look at how they’re connected, there’s no way to split the four cubes into two pairs, each pair connected by its own edge. In other words, you can’t group them into two separate edges without overlap. Therefore, option B is the answer. But let me check the other options just to be sure.
>
> > Correctly identify the structure and explain the reason why it cannot be combined.
>
> Option C: The figure shows a 2x2 square of cubes. Wait, but each block is 2x1x1. So, if you have two of them, you can place them side by side to form a 2x2 square. Yes, that seems possible. For example, place one block vertically on the left and the other vertically on the right. Then you have a 2x2 square. So, option A should be possible.
>
> Option D: Let's see, the backwards L-shape with four cubes. If each block is two cubes, then arranging two blocks to form an L-shape. For example, one block vertical and the other placed horizontally next to it. That would form an L with three cubes, but wait, no. Wait, each block is two cubes. If you place them perpendicular to each other, you can have an L-shape with four cubes. For example, one block is two cubes vertical, and the other block is two cubes horizontal, attached to the vertical block on the right-back side. That would form an L-shape with four cubes. So, that's possible.
>
> Option E: This figure is a 2x2x2 cube. Wait, that would require eight cubes. But Michael only has four cubes. So, he can't make that. Wait, perhaps I miscounted. Let me recount the cubes in option E. The figure is a vertical stack of four cubes. Each block is two cubes, so stacking them would make four cubes. So, that's possible.
>
> > There is a visual hallucination at first, but after reasoning, it begins to doubt whether it makes mistakes and decides to check again. Upon review, it identifies and corrects the error.
>
> Therefore, the answer is **Option B**.
>
> **Answer**: B

---

### Author Response · Authors · 2025-08-09
**Thank you to all the reviewers.**

We would like to express our sincere thanks to all reviewers for your comprehensive and constructive comments on our paper, and are encouraged by your high regard for our efforts during the discussion phase. We are pleased that reviewers appreciate aspects such as the novelty and clear motivation of addressing slow-thinking reasoning in LVLMs, the well-designed SOPHIA framework, and the strong empirical results. We are also eager to seriously consider each reviewer's comments. These questions and suggestions raised during the discussion phase have significantly enhanced our work. We will integrate all the feedback to refine our paper. Thank you once again.

---

### Note · Authors · 2025-08-14

We sincerely thank all reviewers for their active engagement and constructive feedback throughout the discussion. **We have conveyed comments intended for AC's consideration and Reviewer-AC discussion in Paper Clarification part**.

1. In the **Paper Clarification** part, we clarify that SOPHIA leverages alignment of visual and textual embeddings, with carefully designed prompts capturing many low-level cues (color, shape, texture, spatial relations). Results show improvements not only on symbolic tasks but also on perception-related subsets (e.g., MMMU Pro art/history/design, MathVision spatial reasoning). While some detail loss in captioning is inevitable, this limitation is acknowledged and will be discussed.

2. Ablation studies confirm SOPHIA’s semi-off-policy design improves performance, and the comparisons with direct distillation highlight SOPHIA’s unique benefit in on-policy visual understanding.

3. We further conduct the research on Qwen2.5-VL-8B/32B to verify generality. Additionally, manual evaluation of 50 MMMU Pro samples also shows marked reductions in hallucination (38%→16%) and logic errors (54%→44%).

4. Training on InternVL3.0-38B requires ~2048 A800 GPU hours (vs. ~1536 for GRPO); inference latency increases moderately due to longer reasoning, consistent with slow-thinking goals.

All reviewer suggestions, including richer qualitative analysis, broader benchmarks, and clearer methodological exposition, will be integrated. We greatly appreciate the reviewers’ constructive comments which have significantly strengthened the paper.

---

### Decision · Program_Chairs · 2025-09-17

**Decision:**

Accept (poster)

**Comment:**

The paper proposes a semi-off-policy RL method for slow-thinking reasoning in vision language models. The main strength is that it tackles a clear gap in current training strategies, and all reviewers agreed it achieves strong empirical improvements on challenging reasoning benchmarks. Reviewers highlighted the novelty of the semi-off-policy design, the combination of visual grounding with reasoning supervision, and thorough ablations. Concerns included the reliance on caption-based visual representations, limited benchmark diversity, lack of qualitative analysis of error modes, dependence on private training data, and unclear generality across mllm architectures. Reviewers also asked about computational cost and inference latency. The authors responded comprehensively, adding results on Qwen-VL, conducting manual hallucination studies, clarifying training/inference pipeline, and explaining how SOPHIA differs from distillation. They also promised to release internal data. This rebuttal convinced most reviewers, with some increasing their scores and all agreeing the issues were addressed satisfactorily.

The final decision is accept. While the paper is not flawless, e.g., stronger evaluation on perception-heavy tasks and more transparent dataset description would further strengthen it, the method is well-motivated, clearly presented, and achieves high impact results with broad interest. Importantly, the authors engaged carefully with reviewer feedback, expanding analyses and demonstrating generality beyond InternVL. The paper advances the field of mllm reasoning in a meaningful way and sets a solid baseline for future slow-thinking approaches.